# Variation in microbiome and metabolites is associated with advantageous effects of cholestyramine on primary biliary cholangitis with pruritus

Yijun Zhou,[1] Gaoxiang Ying,[1] Wei Shen,[1] Yusheng Cui,[1] Bo Xiang,[1] Muyuan Jiang,[2] Jianfeng Bao,[1] Qiaofei Jin[1]

**ABSTRACT** Emerging evidence implicates bile acid-intestinal microbiota interactions in the pathogenesis of pruritus associated with primary biliary cholangitis (PBC). Cholestyramine, a bile acid sequestrant, is clinically recommended for pruritus alleviation. This study investigates its regulatory effects on gut microbiome composition and metabolite profiles. A prospective cohort of 54 pruritic PBC patients and 25 asymptomatic controls underwent longitudinal multi-omics profiling. Fecal 16S rRNA sequencing and untargeted metabolomics were performed pre-/post-4-week cholestyramine intervention (4 g twice daily). Serum autotaxin, as a biomarker for pruritus assessment, and liver function tests were completed simultaneously. Four important findings were listed as follows. (i) Pruritus phenotype characteristics: Pruritic patients exhibited elevated cholestasis indices (total bilirubin, alkaline phosphatase [ALP], and gamma-glutamyl transferase), higher ATX levels, and increased Gp210 antibody positivity compared to controls (all $P < 0.01$). Cholestyramine significantly reduced 5-D pruritus scores, ATX levels, and cholestasis markers ($P < 0.01$). (ii) Microbial dysbiosis: Gut microbiota diversity (Shannon/Simpson indices) was markedly decreased in pruritic patients, with taxonomic enrichment of *Romboutsia*, *Stenotrophomonas*, and *Achromobacter*, whereas *Lachnospiraceae* and *Bacteroidaceae* predominated in controls. (iii) Metabolomic perturbations: Metabolomic analysis identified diminished medium-chain fatty acids and indole derivatives (e.g., norharman) in pruritic patients. (iv) Therapeutic efficacy: Microbial-metabolite-clinical correlations revealed the pivotal role of the *Romboutsia*-norharman-ATX/ALP axis in the pathogenesis of pruritic PBC. Post-treatment, cholestyramine restored microbial diversity, normalized metabolite levels, and attenuated pruritus. *Enterobacteriaceae*/long-chain fatty acids have been identified as a significant marker for predicting the efficiency of the response to cholestyramine.

**IMPORTANCE** Pruritus in primary biliary cholangitis arises from synergistic cholestasis and gut microbiome-metabolite dysregulation. Cholestyramine mitigates symptoms by modulating the microbiome-metabolite-host axis, highlighting its therapeutic potential through microbiota remodeling and metabolic homeostasis restoration.

**KEYWORDS** primary biliary cholangitis, pruritus, microbiome, metabolites, cholestyramine

P rimary biliary cholangitis (PBC) is an immune-mediated chronic cholestatic liver disease pathologically defined by progressive interlobular bile duct destruction (1). Approximately 75% of patients develop persistent or intermittent pruritus during disease progression, a debilitating symptom that severely compromises quality of life and predisposes to neuropsychiatric complications, including sleep disturbances, anxiety, and depression; severe cases may exhibit suicidal ideation (2). These manifestations

**Peer Reviewers** Djandan Tadum Arthur Vithran, Central South University, Changsha, China; Jianfei Long, Huashan Hospital, Shanghai, China

Address correspondence to Qiaofei Jin, jinqiaofei58095801@163.com.

The authors declare no conflict of interest.

See the funding table on p. 14.

highlight the need to extend therapeutic objectives beyond biochemical parameters to include comprehensive symptom management as a critical clinical endpoint.

Current research faces two key challenges in managing PBC-associated pruritus. First, the pathophysiology remains incompletely understood. While cholestasis is conventionally recognized as the central mediator (3), emerging evidence implicates gut microbiota dysbiosis in pathogenesis through gut-liver axis interactions. Our preliminary work demonstrates significant correlations between intestinal microbial alterations and bile acid metabolic disturbances in PBC patients (4). Mechanistically, gut microbiota regulates bile acid homeostasis through ileal reabsorption processes and bile acid pool composition, while bile acid profiles reciprocally influence microbial community structure (5). This bidirectional crosstalk positions the gut-liver axis as a promising therapeutic target.

Second, objective pharmacodynamic assessment remains limited. Cholestyramine, endorsed as first-line therapy in the 2021 Chinese PBC guidelines (6), primarily reduces systemic bile acid levels through ileal sequestration (7). However, emerging evidence suggests that its efficacy may extend beyond chelation to include gut microbiota remodeling and subsequent bile acid metabolic modulation (8). Clinical evaluation is further complicated by pruritus's subjective nature and the absence of validated response biomarkers.

To address these gaps, we propose a microbiome-metabolite-cholestasis-pruritus axis hypothesis. Our investigation employs multi-omics approaches to characterize biological signatures in pruritic PBC patients, combined with cholestyramine intervention studies to establish objective efficacy biomarkers and predictive therapeutic models. This integrative strategy aims to elucidate molecular mechanisms underlying PBC-associated pruritus while facilitating personalized treatment development.

## MATERIALS AND METHODS

### Study subjects and sample collection

This prospective cohort study consecutively enrolled PBC patients from the inpatient department of Hangzhou Xixi Hospital affiliated to Zhejiang Chinese Medical University between January 2023 and September 2024. A total of 54 pruritic PBC patients with 25 matched non-pruritic PBC patients were enrolled in this study. All participants fulfilled diagnostic criteria per the 2021 Chinese PBC Management Guidelines (6). The cohort was stratified into two groups: a disease group (verbal analog scale [VAS] [9] ≥4) and a non-pruritus control group (VAS score = 0). Pruritus severity was further quantified using the validated 5-D itch scale (10) complemented by the VAS. All participants received ursodeoxycholic acid (UDCA) at standard doses (13–15 mg/kg/day) throughout the study. Exclusion criteria comprised: (i) comorbid malignancies or renal impairment; (ii) pregnancy/lactation status; (iii) recent (8 weeks before enrollment) use of antibiotics, proton pump inhibitors (PPI), or metformin; and (iv) prior cholestyramine exposure.

Cholestyramine was administered twice daily (4 g/dose) before meals, with a mandatory 4-hour separation from other medications to prevent UDCA absorption interference. Biochemical parameters, serum, and stool samples were collected at baseline and 4-week follow-up. One participant withdrew consent due to intolerance to the taste of the cholestyramine.

### Sample preparation

Patients were required to fast overnight before the collection of blood samples on the morning of each visit. Blood samples were centrifuged at 2,800 rpm for 15 minutes at 4°C, and serum was aliquoted and stored at −80°C until analysis. Fecal samples were freshly collected and immediately frozen at −80°C.

## DNA extraction, 16S rRNA gene amplicon sequencing, and data processing

Total genomic DNA was extracted from approximately 200 mg of homogenized fecal material using a commercial DNA isolation kit (QIAamp PowerFecal Pro DNA Kit, Qiagen) following the manufacturer's protocol. DNA purity and concentration were assessed using a NanoDrop spectrophotometer (absorbance ratios at 260/280 nm and 260/230 nm) and quantified via Qubit fluorometry. The hypervariable V3–V4 region of the bacterial 16S rRNA gene was amplified using certain primers. PCR reactions were performed in triplicate with Phusion High-Fidelity DNA Polymerase (Thermo Fisher Scientific). Amplicons were purified using AMPure XP beads (Beckman Coulter) and indexed with unique barcodes for multiplex sequencing. Libraries were pooled in equimolar ratios and sequenced on an Illumina MiSeq platform (Illumina Inc.) using paired-end 2 × 300 bp chemistry, following the manufacturer's guidelines. Raw sequencing reads were quality filtered, trimmed, and clustered into amplicon sequence variants using bioinformatics tools QIIME2 (version 2023.2). Chimeric sequences were removed, and taxonomy was assigned using reference databases SILVA database V.138.

## Fecal metabolome profiling and data preprocessing

Metabolites were extracted using a methanol-water (80:20, [vol/vol]) solution. Metabolite profiling was performed using liquid chromatography-mass spectrometry. The extracted metabolites were separated on a C18 reversed-phase column using a gradient elution with mobile phases consisting of water (0.1% formic acid) and acetonitrile (0.1% formic acid). The mass spectrometer was operated in both positive and negative ionization modes to ensure comprehensive coverage of metabolites. Data were acquired in full-scan mode with a mass range of 50–1,000 m/z. Raw data were converted to mzXML format using ProteoWizard and processed through our in-house R-based analytical pipeline (Biotree, Shanghai) for rigorous quality-controlled peak detection, alignment, and integration. To mitigate false positives inherent in non-targeted metabolomics, metabolite features were excluded if they exhibited >30% relative SD across quality control samples or were detected in <50% of biological samples. Following positive/negative mode feature consolidation, high-confidence annotation of 681 metabolites was achieved (MS2 score >0.8) using our reference library, with this stringently filtered subset utilized for downstream analyses.

## Measurement of serum autotaxin

Serum levels of autotaxin (ATX) were assayed using Human ENPP-2/Autotaxin Quantikine ELISA Kit (Catalog # DENP20, R&D Systems, USA) following the manufacturer's protocol.

## Statistical analysis

Regarding clinical indices, continuous variables were summarized as median (minimum and maximum) for non-parametric data. Categorical variables were reported as frequency (percentage). Analyses were conducted using SPSS 23.0, with a two-tailed $P$-value < 0.05 considered statistically significant. For paired continuous data, the Wilcoxon signed-rank test (a non-parametric test) was utilized when the data deviated from a normal distribution, as verified by the Shapiro-Wilk test. Differences in proportions between groups were evaluated using the $\chi^2$ test.

16S rRNA sequencing data analysis mainly included the following parts. Alpha diversity metrics (Shannon index, Chao1, and observed species) were calculated to assess microbial diversity within samples. Statistical comparisons between groups were performed using non-parametric tests such as the Kruskal-Wallis test or the Wilcoxon rank-sum test. Beta diversity was evaluated using distance metrics (Bray-Curtis, UniFrac) and visualized through principal coordinate analysis (PCoA) or non-metric multidimensional scaling (NMDS). Permutational multivariate analysis of ANOSIM was used to test for significant differences between groups. To identify differentially abundant taxa,

Linear Discriminant Analysis Effect Size (LEfSe) was applied. Gut metabolic profile analysis mainly included the following parts.

The metabolomics data set underwent log2 transformation and variance scaling to standardize variance across multivariate analyses. Prior to fecal metabolome analysis, principal component analysis was performed to identify and exclude outliers beyond the 95% confidence interval threshold. For multivariate modeling, we implemented partial Least Squares Discriminant Analysis (PLS-DA) using SIMCA-P v14.0 (Umetrics AB), which underwent rigorous validation through sevenfold cross-validation, assessment of explained variance ($R^2Y$) and predictive capability ($Q^2$) and permutation testing. Confounder adjustment was implemented via linear mixed-effects modeling, controlling for antibiotic/proton pump inhibitor exposure, ursodeoxycholic acid dosage, and comorbidity status (hepatic/renal impairment) and demographic covariates (age, sex, and body mass index [BMI]). Metabolites retaining Pfdr <0.05 post-adjustment were designated differentially abundant. Stringent curation retained only metabolites with definitive annotations for differential analysis. Selection criteria required metabolites to meet three thresholds: (i) variable importance in projection >1; (ii) fold change magnitude >1 or <−1; and (iii) false discovery rate (FDR)-adjusted $P$-value < 0.05 from Wilcoxon rank-sum testing. Qualified metabolites were subsequently log2-transformed and final identification of differentially abundant metabolites required attainment of $P < 0.05$ in the adjusted linear model analysis.

Correlation analysis between gut microbiota and metabolomic profiles was conducted using a multi-step statistical framework to address high-dimensional data complexity and compositional bias. All analyses were performed in R (v4.3.1) with significance thresholds set at FDR < 0.05 unless otherwise specified. Computed sparse associations using Sparse Spearman correlations (SparCC algorithm) for taxon-metabolite pairs. Constructed microbial-metabolite interaction networks using SPIEC-EASI (SParse InversE Covariance Estimation) with MB neighborhood selection.

## RESULTS

### Clinical characteristics and therapeutic outcomes in cholestyramine-treated pruritic PBC cohorts

A prospective cohort comprising 54 pruritic PBC patients underwent 4-week cholestyramine therapy, with 25 matched non-pruritic PBC patients constituting the control group. One participant withdrew consent due to intolerance to the medication's organoleptic properties. Throughout the study, strict exclusion criteria prohibited concurrent use of antibiotics, PPI, or metformin, with no documented instances of encephalopathy, cholangitis, or gastrointestinal hemorrhage.

Patients with pruritus exhibited significantly elevated markers of cholestatic severity, including serum alkaline phosphatase (ALP) and gamma-glutamyl transferase (GGT) levels (both $P < 0.01$), along with hyperbilirubinemia (serum total bilirubin [TB], $P < 0.01$). They also demonstrated higher autotaxin levels (serum ATX, $P < 0.01$) and increased Gp210 antibody seroprevalence (58% vs 16%, $P < 0.01$) compared to non-pruritic controls.

Following the 4-week cholestyramine regimen, clinical improvements emerged: pruritus severity decreased significantly (5-D score: 20 [15–24] to 7 [5–14], $P < 0.01$) alongside ATX normalization (54.08 [25.35–89.49] to 25.67 [4.35–58.48] ng/mL, $P < 0.01$). Concurrent biochemical enhancements featured marked reductions in ALP, GGT, and TB levels (all $P < 0.01$), collectively demonstrating cholestyramine's multi-parametric therapeutic efficacy (Table 1).

### Distinct gut microbiota alterations in pruritic PBC patients and cholestyramine-induced modulations

Pruritic PBC patients ($n = 54$) exhibited significantly diminished alpha diversity relative to controls ($n = 25$), as quantified through Shannon ($P < 0.01$) and Simpson indices

**TABLE 1** Clinical characteristics of pruritic and non-pruritic PBC patients and therapeutic outcomes after cholestyramine administration

| Characteristics | Group A controls (N = 25) | Group B PBC with pruritus (N = 54) | Group C after cholestyramine administration (N = 53) | P value |
|---|---|---|---|---|
| Age, median age (min, max) | 55 (42–47) | 58 (29–76) | | A vs B: P = 0.50 |
| Female, n (%) | 22 (88) | 48 (89) | | A vs B: P = 0.59 |
| BMI, median (kg/m²; min–max) | 21.27 (19.12–23.56) | 21.47 (17.33–26.67) | | A vs B: P = 0.67 |
| Clinical index, median (min–max) | | | | |
| ALP, U/L | 199 (144–392) | 423 (136–869) | 322 (87–698) | A vs B: P < 0.01B vs C: P < 0.01 |
| GGT, U/L | 150 (103–292) | 474 (216–751) | 404 (155–665) | A vs B: P < 0.01B vs C: P < 0.01 |
| TB, U/L | 14.6 (4.79–45.17) | 70.34 (31.04–99.42) | 54.46 (16.61–86.1) | A vs B: P < 0.01B vs C: P < 0.01 |
| TBA[a], U/L | 39.31 (22.54–58.92) | 72.88 (22.9–129.56) | 52.27 (12.66–119.58) | A vs B: P < 0.01B vs C: P < 0.01 |
| IgM, g/L | 2.62 (0.94–9.13) | 8.1 (2.7–15.3) | 6.8 (1.01–14.57) | A vs B: P < 0.01B vs C: P < 0.05 |
| Pruritus related index, median (min-max) | | | | |
| ATX | 18.69 (11.39–24.29) | 54.08 (25.35–89.49) | 25.67 (4.35–58.48) | A vs B: P < 0.01B vs C: P < 0.01 |
| 5-D score | 0 | 20 (15–24) | 7 (5–14) | B vs C: P < 0.01 |
| Autoantibody positivity, n (%) | | | | |
| AMA-M2 | 25 (100) | 53 (98) | | A vs B: P = 0.69 |
| Sp100 | 10 (40) | 22 (41) | | A vs B: P = 0.58 |
| Gp210 | 4 (16) | 31 (58) | | A vs B: P < 0.01 |

[a]TBA, total bile acids.

(P < 0.01). Notably, cholestyramine administration induced substantial restoration of microbial diversity, with post-treatment metrics approximating control values (Shannon P < 0.01; Simpson P < 0.01; Fig. 1A and B). Beta-diversity analysis via PCoA and NMDS revealed marked structural divergence between pruritic PBC patients and controls (Fig. 1C and D). Intriguingly, therapeutic intervention stratified the microbial architecture into two distinct cohorts: one cluster demonstrating partial restoration toward controls, while the other maintained significant compositional dissimilarity from controls.

## Taxonomic signatures of pruritic PBC and cholestyramine-mediated reconfiguration

LEfSe-derived cladograms revealed distinct gut microbiome distribution patterns between pruritic PBC patients (n = 54, red nodes) and non-pruritic controls (n = 25, green nodes), confirming cohort-specific dysbiosis indices. Subsequent linear discriminant analysis (LDA) at genus resolution identified eight microbial taxa exhibiting diagnostic potential (LDA > 3; P < 0.05), with five genera enriched in pruritic patients, including g_Stenotrophamonas, g_Achiromobacter, o_Burkholderiales, g_Romboutsia, and f_Peptostreptococcaceae (all P < 0.05 and LDA > 3). Control subjects demonstrated characteristic enrichment of f_Lachnospiraceae, f_Bacteroidaceae, and f_Enterobacteriaceae (all P < 0.05 and LDA > 3; (Fig. 2A and B).

Post-cholestyramine intervention elicited significant taxonomic normalization, with complete inversion of pretreatment microbial signatures. Notably, Lachnospiraceae, Bacteroidaceae, and Romboutsia exhibited the most pronounced compositional shifts (all P < 0.05; (Fig. 2C).

## Fecal metabolomic alterations in pruritic PBC and cholestyramine-driven metabolic reconstitution

Untargeted metabolomic profiling of fecal samples (pruritic PBC: n = 49; post-cholestyramine: n = 45; controls: n = 23) elucidated gut-liver axis metabolic perturbations. PLS-DA demonstrated significant metabolic divergence between pruritic PBC patients and controls ($R^2Y$ = 0.91, $Q^2Y$ = 0.81, P < 0.05). Post-intervention stratification revealed bifurcated metabolic states: one cluster exhibiting partial restoration toward control profiles, the other maintaining dysregulated signatures (Fig. 3A).

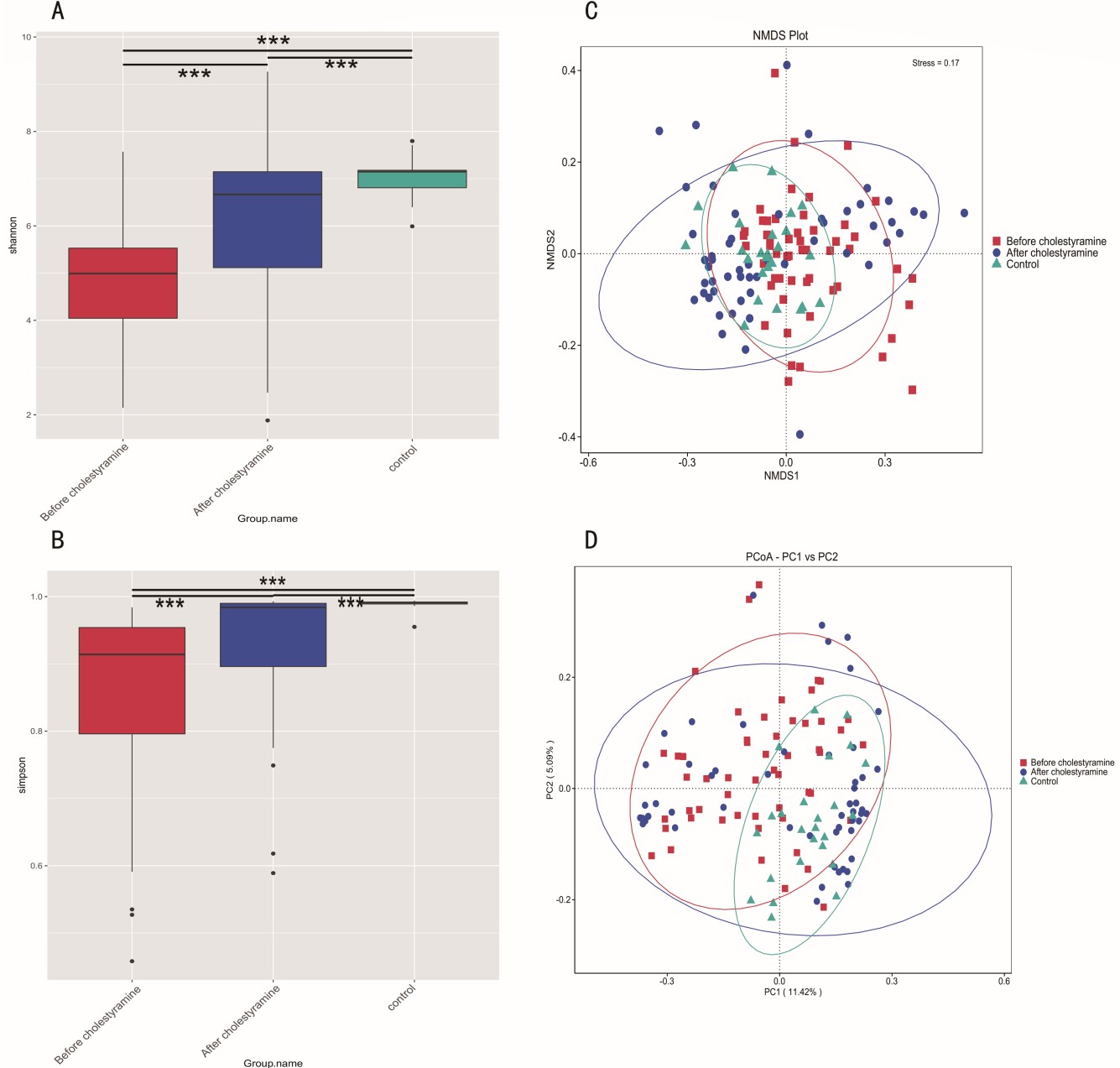

**FIG 1** Gut microbial diversity in pruritic PBC patients and controls, and alterations after cholestyramine administration. (A) Shannon diversity index. \*\*\**P* < 0.001. (B) Simpson diversity index. \*\*\**P* < 0.001. (C) Beta-diversity analysis via PCoA. (D) Beta-diversity analysis via NMDS.

Differential analysis identified 393 ionic features distinguishing pruritic from control patients (140 anions, 253 cations; all *P* < 0.05). Key findings included significant depletion of barrier-enhancing indole derivatives in pruritic cohorts: norharman (log$_2$FC = −0.55), indole-3-butyric acid (log$_2$FC = −0.87), and XLR11 N-(2-fluoropentyl) isomer (log$_2$FC = −1.50; \**P*\* <0.05 for all; Fig. 3B). Cholestyramine treatment effectively restored these AhR-activating tryptophan metabolites to near-normal levels (post-treatment log$_2$FC: 0.60, 1.01, and 3.27, respectively vs. baseline; *P* < 0.05). Concurrently, the antifungal/anti-inflammatory medium-chain fatty acid (MCFA) decanoic acid (initially elevated at log$_2$FC = 2.90) showed significant reduction after intervention (post-treatment log$_2$FC = 1.15, *P* < 0.05; Fig. 3C).

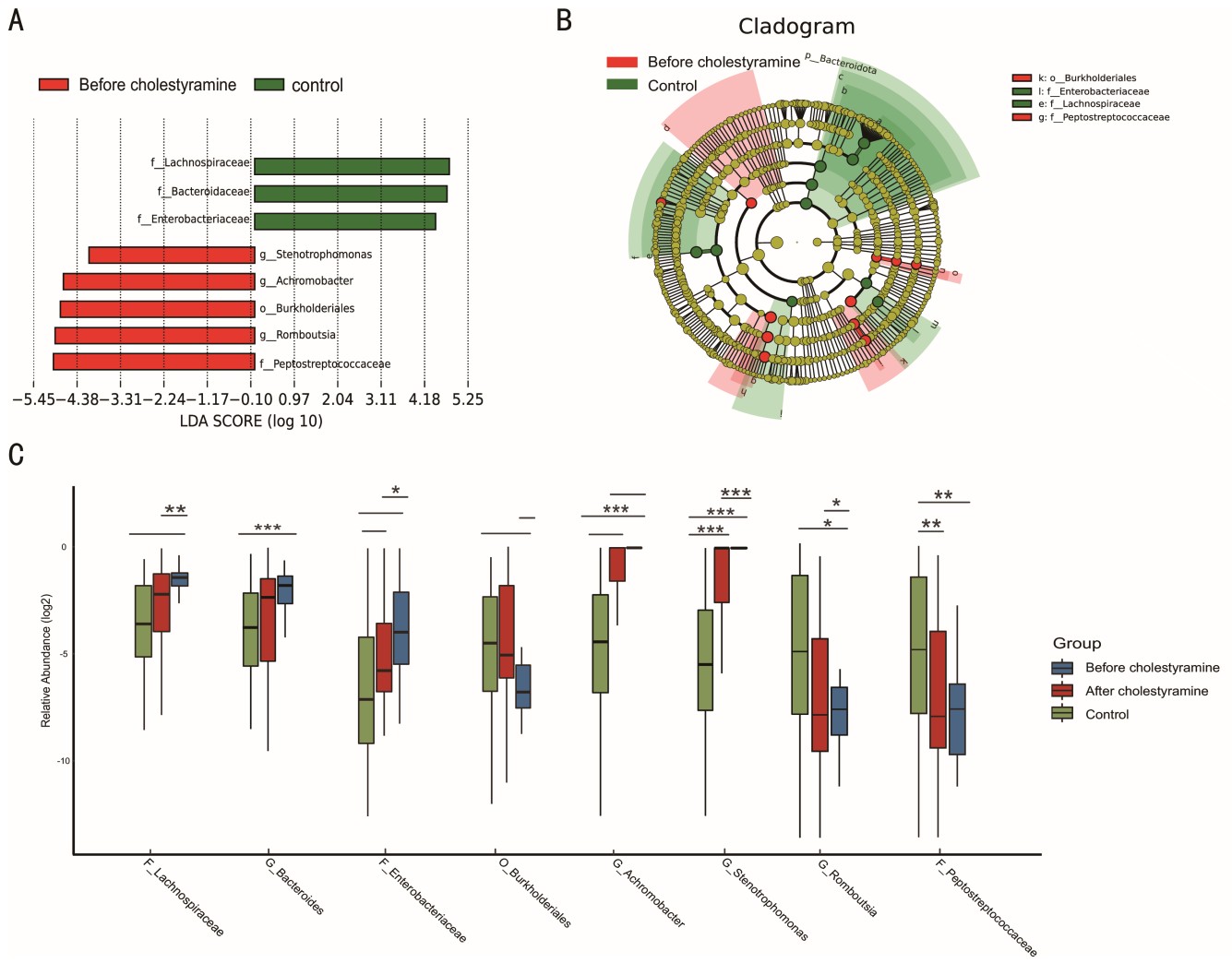

**FIG 2** Variations of fecal microbiota composition in pruritic PBC patients and alterations after cholestyramine administration. (A and B) LDA effect size analysis between pruritic and non-pruritic PBC patients. (C) Characteristic gut microbial alterations after cholestyramine administration. *$P < 0.05$; **$P < 0.01$; ***$P < 0.001$.

## Associations between the disease-linked microbiota and metabolites

Significant covariation emerged between fecal metabolomic profiles and gut microbial composition across the cohort, reinforcing the tight functional coupling between gut microbiota and their metabolic output. Notably, disease-enriched taxa exhibited a pronounced inverse relationship with metabolite profiles enriched in controls (Fig. 4A). Specific negative associations were identified between norharman levels and disease-associated *Peptostreptococcaceae* genera ($r = -0.25$, $P = 0.04$), with additional negative correlations between norharman and *Romboutsia* abundance ($r = -0.25$, $P = 0.03$). Conversely, 10-undecenoic acid (a medium-chain fatty acid) demonstrated positive associations with *Enterobacteriaceae* genera enriched in controls ($r = 0.41$, $P < 0.01$; Fig. 4B).

## Associations of microbial taxa and metabolites with clinical phenotype

To investigate pathophysiological relevance, we performed covariate-adjusted partial Spearman correlations between disease-associated features and clinical parameters. Microbial taxa overrepresented in pruritus-afflicted PBC patients displayed preferential associations with cholestasis markers (TB and ALP), whereas metabolites showed stronger links with ATX levels, a pruritus biomarker (Fig. 5A). Elevated *Romboutsia*

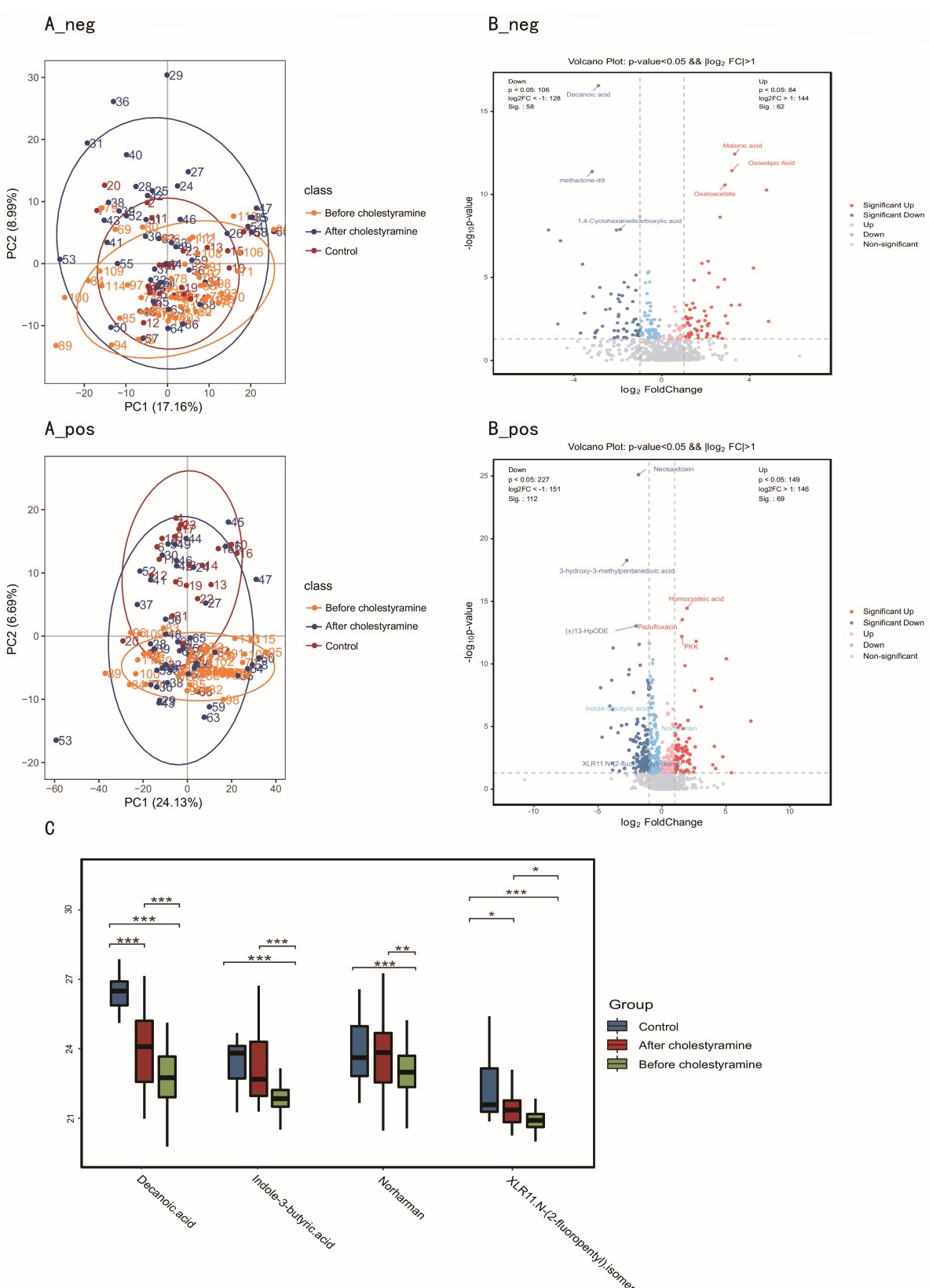

**FIG 3** Fecal metabolome changes in pruritic PBC vs controls, and alterations after cholestyramine administration. (A) PCoA in controls, pruritic PBC, and after cholestyramine administration (A_neg; A_pos). (B) Volcano plot demonstrated metabolite changes in pruritic PBC vs controls (B_neg; B_pos). (C) Box plot showed four characteristic metabolome changes after cholestyramine administration. *$P < 0.05$; **$P < 0.01$; ***$P < 0.001$.

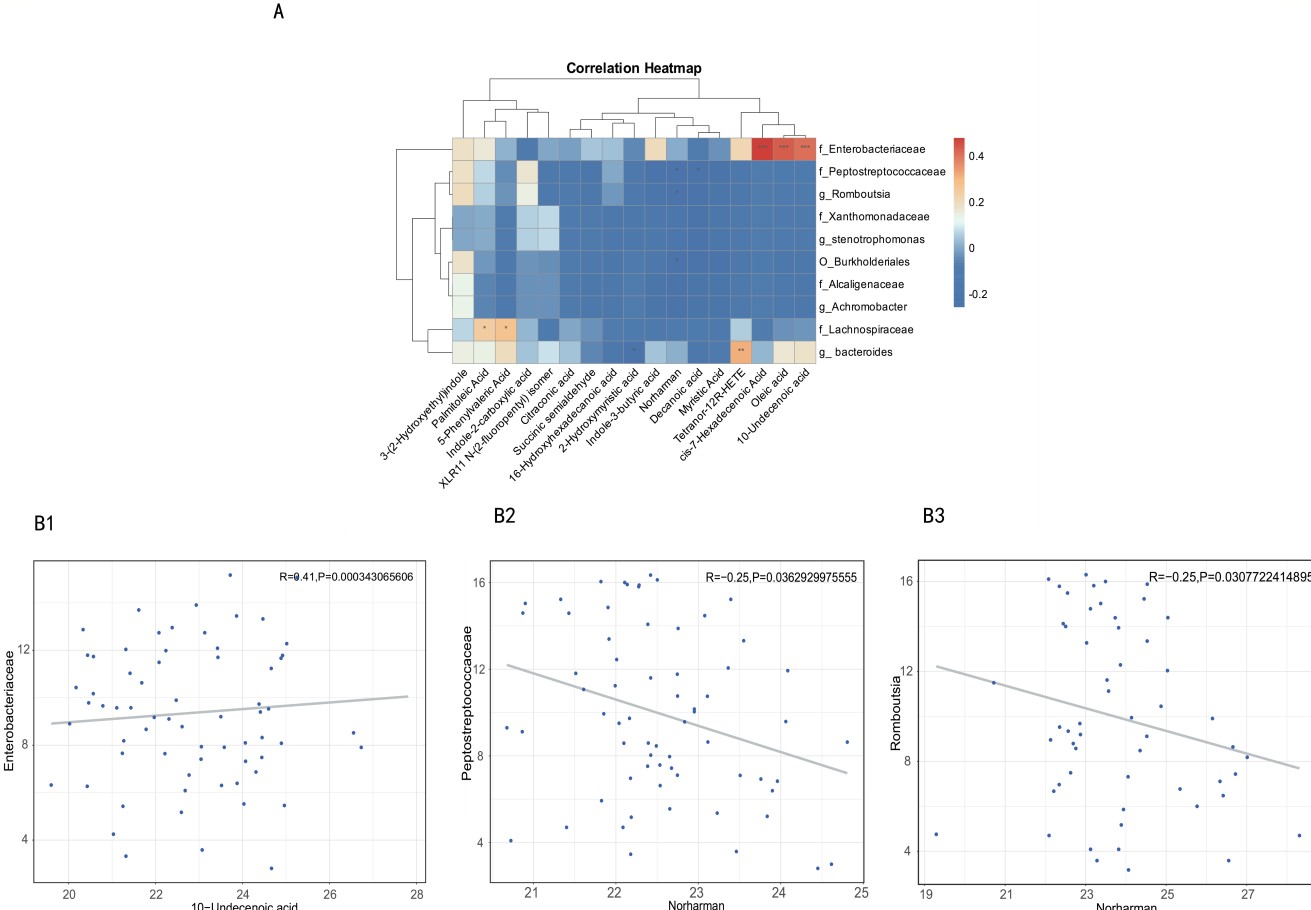

**FIG 4** Associations of disease-related taxa and metabolites. (A) The heatmap depicts relationships between the taxa and metabolites changed in pruritic PBC and controls. *$P < 0.05$; **$P < 0.01$; ***$P < 0.001$. (B) Examples of individual taxa-metabolite associations. Abundances of taxa and metabolites are plotted after log2 transformation, and 0 values were assigned 1e-05. Each dot represents one sample. (B1 10-Undecenoic acid- *Enterobacteriaceae*; B2 Norharman-*Peptostreptococcaceae*; B3 Norharman-*Romboutsia*).

abundance correlated with increased cholestasis markers (ALP: $r = 0.36$, $P < 0.01$; GGT: $r = 0.31$, $P < 0.01$; TB: $r = 0.31$, $P < 0.01$; total bile acids: $r = 0.36$, $P < 0.01$). Intriguingly, diminished circulating levels of norharman paralleled increased ATX ($r = -0.42$, $P < 0.01$) and ALP ($r = -0.42$, $P < 0.01$; Fig. 5B).

## Microbiome-metabolome axis modulates therapeutic heterogeneity in cholestyramine response

Multivariate analysis revealed dichotomous microbial and metabolic stratification: one cluster exhibited partial restoration toward control-state configurations, while the other maintained marked compositional divergence, indicative of differential therapeutic efficacy of cholestyramine. Subjects were classified using the 5-D pruritus score reduction threshold (>50% = superior responders [SR]; <50% = inferior responders [IR]). Thirty-seven subjects were classified as SR, while 16 subjects were classified as IR. Comparative metagenomic profiling identified *Enterobacteriaceae* as taxonomically enriched in the IR cohort ($P < 0.05$, LDA > 3; Fig. 6A). Parallel metabolomic analyses demonstrated IR-specific accumulation of long-chain fatty acids (LCFAs), notably 11(E)-eicosenoic acid (|$\log_2FC$| = 4.46, $P < 0.05$), palmitic acid (|$\log_2FC$| = 3.20, $P < 0.05$), and arachidic acid (|$\log_2FC$| = 2.90, $P < 0.05$; Fig. 6B).

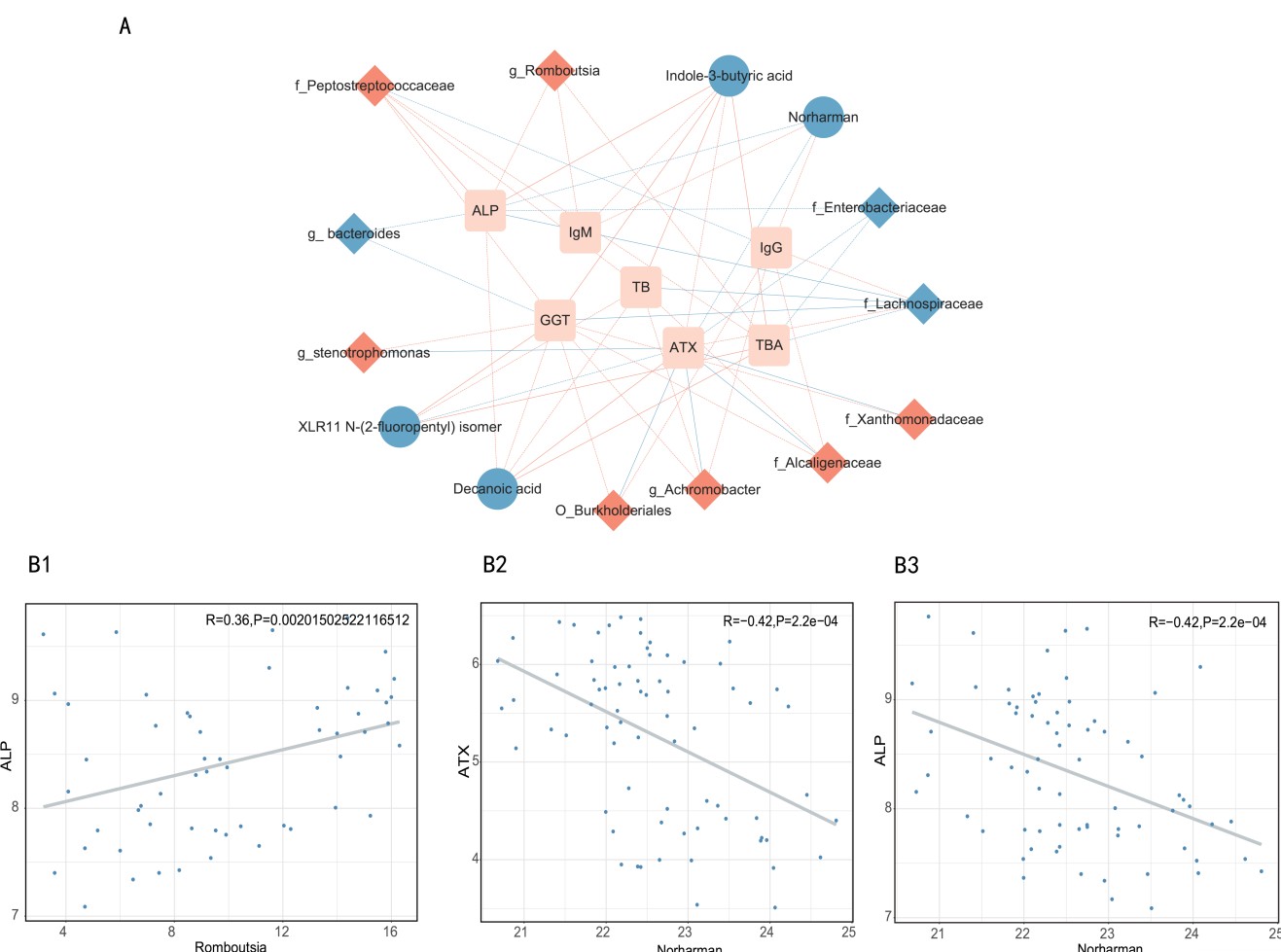

**FIG 5** Integrative network of associations reflecting host-microbe interactions. (A) Network revealed both significant (*P* < 0.05) and suggestive associations (*P* < 0.05 and |ρ| > 0.3, partial Spearman analysis) between differentially abundant taxa or metabolites and clinical indexes in pruritic PBC patients. Nodes represent features increased (in red) or decreased (in blue) in disease groups compared with controls. Lines connecting nodes indicate positive (red) or negative (blue) correlations. The solid and dashed lines denote significant and suggestive correlations, respectively. (B) Examples of individual taxa/metabolite-clinical index associations. Abundances of taxa and metabolites are plotted after log2 transformation, and each dot represents one sample. B1 *Romboutsia*: ALP; B2 Norharman: ATX; B3 Norharman: ALP. TBA, total bile acids; IgG, immunoglobulin G; IgM, immunoglobulin M.

## DISCUSSION

While prior landmark work by Li et al. (8) linked cholestyramine's benefits in icteric PBC patients to compositional and functional alterations in gut commensals, our study pioneers the integration of gut microbiome, metabolome, and clinical indices to characterize moderate-to-severe pruritus in PBC—a distinct clinical phenotype with unique therapeutic challenges. To our knowledge, this represents the first comprehensive analysis of cholestyramine's impact on gut microbial ecology and associated metabolic pathways specifically in pruritic PBC patients. Results showed that pruritic PBC patients display significant gut dysbiosis, as indicated by reduced alpha-diversity, concurrent alterations in community structure and taxon abundance, and a disease-specific increase in pro-inflammatory genera (e.g., *Romboutsia* [11]) coupled with a decrease in commensal taxa. Additionally, these patients exhibit perturbations in their metabolome, particularly systemic decreases in certain metabolites, such as MCFAs and tryptophan-derived indoles (e.g., Norharman [12]). Interestingly, Norharman levels were inversely correlated with the abundance of disease-associated *Romboutsia*. Intervention

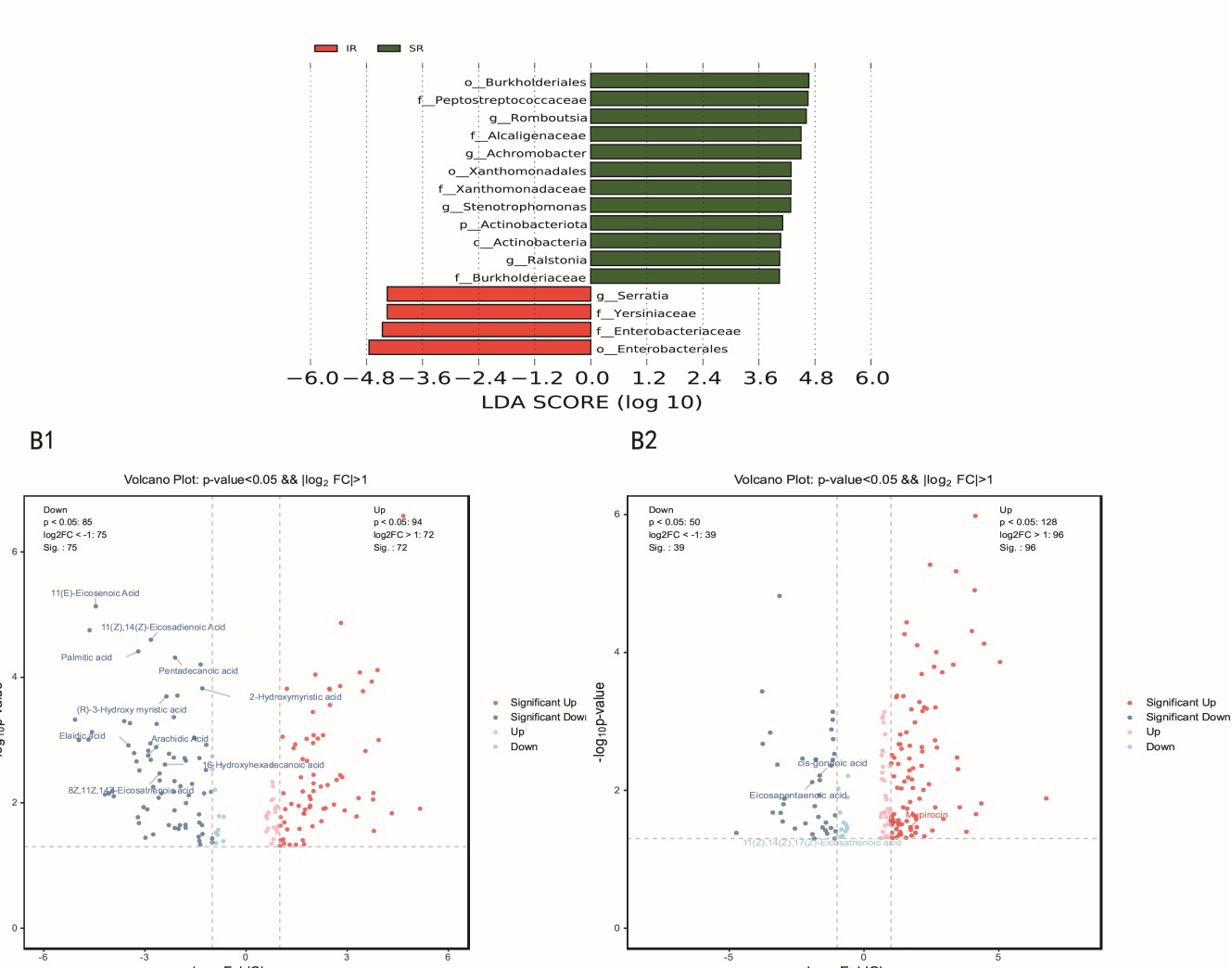

**FIG 6** Distinct changes of microbiota and metabolites between SR and IR. (A) LDA effect size analysis between SR and IR groups. (B) Volcano plot demonstrated metabolite changes in SR vs IR.

analysis revealed cholestyramine's dual mechanism of action, involving microbiome remodeling and metabolite restoration. These coordinated changes were associated with a reduction in clinical pruritus following cholestyramine administration.

16S rRNA sequencing analysis revealed three downregulated and five upregulated genera in pruritic PBC patients, with three genera demonstrating particular pathophysiological relevance.

### *Lachnospiraceae* depletion

This butyrate-producing family showed marked downregulation in pruritic PBC patients. *Lachnospiraceae*-derived short-chain fatty acids (13), particularly acetate and butyrate, play crucial roles in intestinal homeostasis through multiple mechanisms (14). Butyrate enhances intestinal barrier integrity by regulating tight junction proteins and upregulating mucin-2 expression to fortify the mucus layer against pathogens (15). The observed depletion suggests compromised gut barrier function in pruritic PBC patients compared to non-pruritic counterparts.

## *Bacteroides* suppression

The reduced *Bacteroides* abundance likely impairs bile salt hydrolase (BSH) activity, given this genus's established role in bile acid metabolism (16). BSH-mediated deconjugation, predominantly facilitated by *Lactobacillus* (17), *Bifidobacterium* (18), *Clostridium* (19), and *Bacteroides* (20), is critical for secondary bile acid formation. This deficiency can disrupt bile acid homeostasis, potentially exacerbating cholestasis through impaired conversion of primary to secondary bile acids.

## *Romboutsia* elevation

The upregulated *Romboutsia* correlated with dyslipidemia markers, including elevated total cholesterol (TC) and triglycerides (21). Clinical evidence associates TC levels ≥200 mg/dL with disrupted glycerophospholipid-sphingolipid metabolism and poorer PBC prognosis (22), suggesting *Romboutsia* overgrowth reflects upstream cholesterol metabolism impairment secondary to severe cholestasis. Notably, cholestyramine intervention reversed these microbial alterations. This suggests that its therapeutic effects may involve gut microbiota modulation, which coincides with three key physiological changes: intestinal barrier restoration, enhanced bile acid metabolism, and reduced TC levels. Together, these correlated changes could serve as potential biomarkers for managing PBC pruritus while offering insights into cholestyramine's efficacy mechanisms.

Integrated spectrometry-chromatography analyses identified two principal classes of dysregulated metabolites in pruritic PBC patients.

## Indole pathway dysregulation

Significant downregulation of indole derivatives, including indole-3-butyric acid, norharman, and XLR11 N-(2-fluoropentyl) isome, was observed. Tryptophan, the primary indole precursor (23), undergoes microbial conversion in the colon via tryptophanase-expressing species (24, 25), including *Escherichia coli* (26), *Clostridium* (27), *Bacteroides* (28), and *Proteus vulgaris* (29). Impaired tryptophan metabolism leads to the accumulation of two neuroactive intermediates, kynurenic acid and quinolinic acid (30). Kynurenic acid activates transient receptor potential V1 (31) and transient receptor potential A1 channels on cutaneous sensory neurons (32), stimulating pruritogenic neurotransmitter release (33). Whereas quinolinic acid potentiates N-methyl-D-aspartate receptor-mediated neuronal hyperexcitability (34) and substance *P* release (35). Concurrently, diminished Norharman, a *Lactobacillus*-derived histone deacetylase (HDAC1-4) inhibitor, reduces Rftn1-mediated suppression of macrophage M1 polarization (36), exacerbating inflammation. Indole deficiency further compromises intestinal barrier integrity by reducing mucin production (37). This dual imbalance, characterized by reduced immunomodulatory indoles and elevated neurostimulatory metabolites, may contribute to establishing a pruritogenic feedback loop that potentially aggravates cholestatic inflammation.

## MCFA depletion

MCFA downregulation impairs bile acid homeostasis through reducing fecal bile acid excretion via ileal bile acid binding protein suppression (38) and disrupting cholesterol catabolism (39). This metabolic shift perpetuates hypercholesterolemia, a recognized prognostic marker in PBC (40). Cholestyramine administration reversed key metabolic perturbations, coinciding with the restoration of indole and MCFA levels. This metabolic shift occurred alongside enhanced bile acid/cholesterol clearance and attenuation of neuroimmune pruritus pathways—collectively associating with pruritus alleviation. Importantly, restored indole levels may support intestinal barrier integrity, while normalized MCFAs could potentially mitigate cholestatic hypercholesterolemia.

Based on the above findings, we attempted to establish the relationship between clinical biomarkers and gut microbiota and metabolites. ATX, a key enzymatic mediator

of cholestatic pruritus (41), demonstrates specific involvement in pruritogenesis through its catalytic conversion of lysophosphatidylcholine to lysophosphatidic acid (LPA) (42). This biochemical pathway has established ATX activity monitoring as the current satisfied standard for assessing cholestasis-associated pruritus. However, significant diagnostic limitations persist due to ATX's nonspecific elevation across diverse pathological states, including neoplastic disorders (e.g., breast cancer [43]), autoimmune conditions (e.g., systemic lupus erythematosus [44]), and metabolic dysregulation (45). This diagnostic ambiguity underscores the critical need for developing more specific biomarkers in pruritus evaluation. Our integrative analysis reveals a novel pathophysiological axis in PBC-associated pruritus: the *Romboutsia*-Norharman-ATX/ALP triad. Mechanistically, cholestyramine's therapeutic effects likely operate through bile acid sequestration in the ileum, which simultaneously: (i) alters gut microbial ecology: suppresses *Romboutsia* overgrowth while promoting commensal bacteria that produce Norharman; (ii) modulates bile acid metabolism: reduces hydrophobic bile acid burden and enhances farnesoid X receptor/fibroblast growth factor 19 signaling (46), thereby decreasing hepatic ALP synthesis; and (iii) attenuates neuroimmune activation: Norharman enrichment may inhibit ATX expression via serotonin receptor modulation, disrupting the LPA-mediated pruritogenic cascade. Clinically, these interconnected mechanisms correlate with reduced cholestatic severity markers (ALP), diminished ATX activity, and consequent pruritus alleviation after cholestyramine intervention.

Pre-therapeutic microbial and metabolomic profiling demonstrated significant associations with cholestyramine therapeutic efficacy in pruritus management. Non-responders exhibited baseline enrichment of *Enterobacteriaceae* and elevated LCFAs, suggesting microbial-derived predictors of pharmacological resistance. Mechanistically, *Enterobacteriaceae*-sourced lipoprotein (LPS) promotes dual-barrier disruption through toll-like receptor 4-dependent occludin suppression in intestinal epithelium (47) and NKT cell-mediated biliary injury (48). Concurrently, saturated LCFAs potentiate endotoxin effects via NLRP3 inflammasome activation (49) while impairing farnesoid X receptor-regulated mucosal repair pathways (50). These findings position intestinal barrier integrity metrics as stratification biomarkers for cholestyramine response. Implementing baseline *Enterobacteriaceae*/LCFA quantification could enable early escalation to second-line agents (51) (sertraline/rifampicin) in predicted non-responders and microbial reprogramming strategies to resensitize refractory cases.

Furthermore, this study has several limitations that warrant consideration. First, the exclusive recruitment of PBC patients from a single center (Hangzhou Xixi Hospital) limits geographic and ethnic diversity, potentially introducing selection bias that restricts generalizability. Future validation through multi-center cohorts encompassing diverse regions and ethnicities is essential to confirm both baseline gut microbiota/metabolite profiles in pruritic PBC patients and cholestyramine-induced therapeutic modifications. Second, we must emphasize the inherent challenge in establishing causal relationships from our observational study design, particularly given the multifactorial pathophysiology of PBC-associated pruritus, where neurological, immunological, and metabolic pathways interact. While our multi-omics analyses revealed compelling associations between cholestyramine-induced metabolite shifts and pruritus alleviation, these findings cannot definitively attribute causality due to the intricate feedback loops within microbiome-metabolite-host interactions and potential confounding from unmeasured variables. This interpretive limitation is compounded by the absence of functional validation through *in vitro* systems. Future mechanistic studies employing enterocyte/organoid models in controlled experimental settings will be essential to disentangle these complex relationships and establish directional causality between specific microbial metabolic functions and clinical outcomes. Third, this study acknowledges methodological constraints inherent to 16S rRNA sequencing. Although LEfSe analysis identified differentially abundant taxa potentially relevant to PBC pruritus, their specific metabolic pathways and mechanistic roles require further validation. Future

investigations would ideally employ metagenomic approaches to elucidate functional gene networks, subject to resource availability.

In conclusion, PBC-associated pruritus arises from gut-liver axis disruption involving cholestasis-aggravated dysbiosis and microbial tryptophan metabolism hijacking. Cholestyramine exerts dual therapeutic mechanisms via microbiota remodeling and metabolic restoration. Baseline *Enterobacteriaceae*/LCFA profiling enables precision therapy stratification. These findings highlight the therapeutic potential of targeting gut-derived microbial metabolites as a promising adjunctive approach for cholestatic pruritus management.

## ACKNOWLEDGMENTS

We sincerely thank all the participants, working staff, and researchers for their valuable contributions to the study. Importantly, we extend our gratitude to Professor Xiong Ma from Renji Hospital, School of Medicine, Shanghai Jiao Tong University, for his invaluable assistance and support in the manuscript.

This study was funded by grants from the Health Commission of Zhejiang Province (No. 2023KY197 and No. 2025KY147) and Hangzhou Municipal Health and Family Planning Commission (No. 2025HZGF09).

Y.Z. and Q.J. developed the study concept. J.B. and B.X. supervised the project. Y.Z., G.Y., W.S., and Y.C. collected the clinical data and conducted the experiments. Y.Z. and G.Y. assessed the clinical response. M.J. contributed to the pruritus assessment. Y.Z. and G.Y. performed the data analysis and wrote the manuscript with the help of Q.J. and J.B. All authors read and approved the final version of the manuscript. Y.Z.: formal analysis, methodology, visualization, writing—original draft, funding acquisition, and project administration. G.Y.: formal analysis, investigation, methodology, resources, and writing—original draft. W.S.: validation and writing—original draft. Y.C.: data curation and writing—original draft. M.J.: data curation and writing—original draft. B.X.: conceptualization and writing—original draft. Q.J.: investigation, methodology, resources, validation, writing—original draft, and writing—review and editing. J.B.: conceptualization, methodology, resources, writing—original draft, and writing—review and editing.

## AUTHOR AFFILIATIONS

[1]Department of Hepatology, Hangzhou Xixi Hospital affiliated to Zhejiang Chinese Medical University, Hangzhou, Zhejiang, China
[2]Department of Dermatology, Zhejiang Province Hospital of Integrated Traditional Chinese and Western Medicine, Hangzhou, Zhejiang, China

## AUTHOR ORCIDs

Yijun Zhou  https://orcid.org/0000-0001-8607-4999
Qiaofei Jin  http://orcid.org/0009-0007-8738-8982

## FUNDING

| Funder | Grant(s) | Author(s) |
| --- | --- | --- |
| Health Commission of Zhejiang Province | 2023KY197, 2025KY147 | Yijun Zhou |
| Hangzhou Municipal Health and Family Planning Commission | 2025HZGF09 | Jianfeng Bao |

## AUTHOR CONTRIBUTIONS

Yijun Zhou, Formal analysis, Funding acquisition, Investigation, Project administration, Writing – original draft | Gaoxiang Ying, Conceptualization, Data curation, Formal

analysis, Methodology, Validation, Visualization | Wei Shen, Data curation, Methodology, Visualization, Writing – original draft | Yusheng Cui, Formal analysis, Methodology, Supervision, Validation, Writing – original draft | Bo Xiang, Conceptualization, Supervision, Visualization, Writing – review and editing | Muyuan Jiang, Data curation, Supervision, Validation, Writing – review and editing | Jianfeng Bao, Conceptualization, Supervision, Validation, Visualization, Writing – review and editing | Qiaofei Jin, Conceptualization, Methodology, Project administration, Supervision, Validation, Writing – review and editing

## DATA AVAILABILITY

The data sets presented in this study can be found in online repositories. The 16S rRNA sequencing data are available in the NCBI Sequence Read Archive under accession numbers PRJNA1238545 and PRJNA892581 (https://www.ncbi.nlm.nih.gov/), while the metabolomic data can be accessed through the Metabolomics Workbench via https://doi.org/10.21228/M85Z50 (https://www.metabolomicsworkbench.org).

## ETHICS APPROVAL

This study was carried out in compliance with the Helsinki Declaration and received ethical approval from the Ethics Committee of Hangzhou Xixi Hospital affiliated to Zhejiang Chinese Medical University (No. 2022-72). Written informed consent was obtained from each patient.

## ADDITIONAL FILES

The following material is available online.

### Supplemental Material

**Supplemental material (Spectrum00747-25-s0001.xlsx).** Clinical data for patients.

### Open Peer Review

**PEER REVIEW HISTORY (review-history.pdf).** An accounting of the reviewer comments and feedback.

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
