## [Reviewer comments · Microbiology Spectrum]

Microbiology Spectrum

Variation in microbiome and metabolites are associated with advantageous effects of cholestyramine on primary biliary cholangitis with pruritus

Yijun Zhou, Gaoxiang Ying, Wei Shen, Yusheng Cui, Bo Xiang, Muyuan Jiang, Jianfeng Bao, and Qiaofei Jin

Corresponding Author(s): Qiaofei Jin, Xixi Hospital of Hangzhou

Review Timeline:

Submission Date:	March 12, 2025
Editorial Decision:	June 9, 2025
Revision Received:	June 23, 2025
Editorial Decision:	July 22, 2025
Revision Received:	August 22, 2025
Accepted:	October 15, 2025

Editor: Qi Su

Reviewer(s): Disclosure of reviewer identity is with reference to reviewer comments included in decision letter(s). The following individuals involved in review of your submission have agreed to reveal their identity: Djandan Tadum Arthur Vithran (Reviewer #1); LONG Jianfei (Reviewer #2)

Transaction Report:

DOI: <https://doi.org/10.1128/spectrum.00747-25>

Re: Spectrum00747-25 (**Variation in microbiome and metabolites are associated with advantageous effects of cholestyramine on primary biliary cholangitis with pruritus**)

Dear Dr. Qiaofei Jin:

Thank you for the privilege of reviewing your work. Below you will find my comments, instructions from the Spectrum editorial office, and the reviewer comments.

Revision Guidelines

Sincerely,
Qi Su
Editor
Microbiology Spectrum

Reviewer #1 (Comments for the Author):

Comments and Suggestions for the Author:

1.Areas for Improvement:

The discussion sometimes overstates causality (e.g., about gut barrier restoration and MCFA effects); tempering these

statements to reflect correlative rather than causal findings would enhance scientific rigor.

Figures: Some figures (especially network graphs) have small fonts and dense layouts; improving clarity and legend explanations would greatly help readers.

Language polishing: Minor issues like long sentences, passive voice, and redundancy (especially in the Background) should be professionally edited.

Data availability: While it is stated that data are available upon request, depositing sequencing and metabolomics datasets into public repositories (such as NCBI SRA, EBI) would improve transparency and reproducibility.

2. Specific Minor Suggestions:

Page 7: Clarify the sample size after dropout; 54 → 53 in some parts - be consistent.

Page 18: When discussing norharman restoration, specify it as a potential mediator of pruritus alleviation rather than confirmed.

Tables and figures should include explanations for all abbreviations (e.g., ATX, ALP, LCFA) directly in figure legends.

3. Suggestions for Future Research:

Validation cohorts from different geographic regions or ethnicities would significantly enhance generalizability.

Functional assays (e.g., cell culture models) could validate the inferred microbiota-metabolite-host interactions.

Investigation of targeted microbial therapy (e.g., Norharman supplementation, microbiota modulation) as adjunctive treatment in cholestatic pruritus.

Reviewer #2 (Comments for the Author):

This study utilized a multi-omics approach to explore the pathogenesis of pruritus in PBC patients, identifying alterations in the gut microbiome and metabolites. It also demonstrated that cholestyramine effectively alleviates pruritus by modulating the gut microbiome and metabolites, and proposed the mechanism involving the gut microbiome-metabolite-cholestasis-pruritus axis. While the study indicates that cholestyramine may alleviate PBC pruritus through modulating the gut microbiome-metabolite-host axis, the design was observational, which precludes definitive conclusions about causality. Although a cholestyramine intervention was conducted, other factors might have concurrently influenced the gut microbiome, metabolites, and pruritus symptoms, so the observed relationships could be correlative rather than causal.

111. Gut microbiome data

Sample size and diversity representation: The limited number of PBC patients, all recruited from a single center (Hangzhou Xixi Hospital affiliated with Zhejiang Chinese Medical University), may introduce selection bias. This restricts the generalizability of the findings to all PBC patients, as gut microbiome diversity can vary significantly with region, race, and lifestyle.

Depth of microbiome identification and functional analysis: 16S rRNA sequencing only reveals microbial community composition. Our understanding of specific microbial functions, metabolic pathways, and interactions remains limited. While LEfSe analysis identified differential microbes, their exact pathogenic roles in PBC pruritus, whether promotive or inhibitory, were not validated.

2. Metabolome data

Ambiguity in causality of metabolite changes: Although certain metabolite levels changed after cholestyramine treatment, it is unclear whether these metabolites are a cause or a consequence of pruritus. For instance, the study notes associations between indole derivatives, medium-chain fatty acids, and pruritus, but the underlying mechanisms and directional relationships remain unexplored.

Integrity and accuracy of metabolome data: Non-targeted metabolomics may produce noisy data with false positives. Errors can be introduced during data preprocessing steps such as peak detection and alignment, affecting subsequent analysis.

Additionally, not all differential metabolites were structurally identified and quantified, increasing the risk of annotation errors.

3. "received ethical was approved by..." contains spelling and grammatical errors. It should be corrected to "received ethical approval from..."

This study utilized a multi-omics approach to explore the pathogenesis of pruritus in PBC patients, identifying alterations in the gut microbiome and metabolites. It also demonstrated that cholestyramine effectively alleviates pruritus by modulating the gut microbiome and metabolites, and proposed the mechanism involving the gut microbiome-metabolite-cholestasis-pruritus axis.

While the study indicates that cholestyramine may alleviate PBC pruritus through modulating the gut microbiome-metabolite-host axis, the design was observational, which precludes definitive conclusions about causality. Although a cholestyramine intervention was conducted, other factors might have concurrently influenced the gut microbiome, metabolites, and pruritus symptoms, so the observed relationships could be correlative rather than causal.

111. Gut microbiome data

Sample size and diversity representation: The limited number of PBC patients, all recruited from a single center (Hangzhou Xixi Hospital affiliated with Zhejiang Chinese Medical University), may introduce selection bias. This restricts the generalizability of the findings to all PBC patients, as gut microbiome diversity can vary significantly with region, race, and lifestyle.

Depth of microbiome identification and functional analysis: 16S rRNA sequencing only reveals microbial community composition. Our understanding of specific microbial functions, metabolic pathways, and interactions remains limited. While LEfSe analysis identified differential microbes, their exact pathogenic roles in PBC pruritus, whether promotive or inhibitory, were not validated.

2. Metabolome data

Ambiguity in causality of metabolite changes: Although certain metabolite levels changed after cholestyramine treatment, it is unclear whether these metabolites are a cause or a consequence of pruritus. For instance, the study notes associations between indole derivatives, medium-chain fatty acids, and pruritus, but the underlying mechanisms and directional relationships remain unexplored.

Integrity and accuracy of metabolome data: Non-targeted metabolomics may produce noisy data with false positives. Errors can be introduced during data preprocessing steps such as peak detection and alignment, affecting subsequent analysis. Additionally, not all differential metabolites were structurally identified and quantified, increasing the risk of annotation errors.

"received ethical was approved by..." contains spelling and grammatical errors. It should be corrected to "received ethical approval from...".

Dear editor,

Thank you very much for your constructive comments and suggestions on our manuscript (Manuscript ID: Spectrum00747-25), entitled “Variation in microbiome and metabolites are associated with advantageous effects of cholestyramine on primary biliary cholangitis with pruritus”. We have revised the manuscript accordingly, and all amendments are indicated by red font in the marked-up manuscript. In addition, our point-by-point responses to the comments are listed below this letter.

Response to Reviewer 1

1.Areas for Improvement:

The discussion sometimes overstates causality (e.g., about gut barrier restoration and MCFA effects); tempering these statements to reflect correlative rather than causal findings would enhance scientific rigor.

Response: In the revised discussion, definitive causal claims have been systematically replaced with correlative interpretations. Specifically:

(1) Cholestyramine-associated alterations in *Lachnospiraceae*, *Bacteroides*, and *Romboutsia* abundance **correlate with** concurrent improvements in intestinal barrier integrity, bile acid metabolism, and TC reduction (Page18, Lines 5-11);

(2) Metabolic perturbation resolution **coincides with** indole/MCFA restoration alongside enhanced bile acid/cholesterol clearance and

neuroimmune pathway attenuation, temporally **associated with** pruritus alleviation (Page 19, Lines 17-22; Page 20, Line 1);

(3) Cholestyramine intervention **correlates with** suppression of Romboutsia overgrowth and Norharman enrichment, paralleling reductions in ALP, cholestatic severity, ATX expression, and pruritus symptoms (Page 20, Line 18-22; Page 21, Line 1).

All mechanistic interpretations now explicitly reflect observational associations from our clinical dataset rather than inferred causality.

Figures: Some figures (especially network graphs) have small fonts and dense layouts; improving clarity and legend explanations would greatly help readers.

Response: All figures have been systematically revised to address the reviewer's concerns regarding clarity and interpretability. These modifications ensure optimal readability while maintaining scientific precision across all graphical elements.

Language polishing: Minor issues like long sentences, passive voice, and redundancy (especially in the Background) should be professionally edited.

Response: The Background section (Page 3-4) has undergone comprehensive language editing by professional academic professor to address all noted issues. This included restructuring lengthy sentences, converting passive constructions to active voice, and eliminating

redundant expressions, resulting in enhanced clarity and grammatical precision throughout the manuscript.

Data availability: While it is stated that data are available upon request, depositing sequencing and metabolomics datasets into public repositories (such as NCBI SRA, EBI) would improve transparency and reproducibility.

Response: The data sets presented in this study can be found in online repositories. The names of the repository / repositories and accession numbers are as follows: <https://www.ncbi.nlm.nih.gov/>, PRJNA1238545 and PRJNA892581.

2. Specific Minor Suggestions:

Page 7: Clarify the sample size after dropout; 54 → 53 in some parts - be consistent.

Response: In the Methods section (Study Subjects and Sample Collection), the sample size has been explicitly clarified as follows: This study initially enrolled 54 pruritic PBC patients alongside 25 matched non-pruritic controls (Page 5, Lines 4-5). One pruritic participant withdrew due to cholestyramine taste intolerance (Page 5, Lines 19-21), resulting in 53 pruritic patients completing the study. All subsequent analyses reflect this final cohort.

Page 18: When discussing norharman restoration, specify it as a potential mediator of pruritus alleviation rather than confirmed.

Response: Related part has been revised as follows. Norharman, as a potential mediator of pruritus, may contribute to establishing a pruritogenic feedback loop that potentially aggravates cholestatic inflammation (Page 19, Lines 9-12).

Tables and figures should include explanations for all abbreviations (e.g., ATX, ALP, LCFA) directly in figure legends.

Response: All abbreviations in tables and figures have been explained in figure legends.

3. Suggestions for Future Research:

Validation cohorts from different geographic regions or ethnicities would significantly enhance generalizability.

Response: We fully acknowledge the reviewer's valid concern regarding generalizability. As explicitly stated in the Limitations section (Page 21, Lines 19-22; Page 22, Lines 1-3), our exclusive recruitment from a single center in Hangzhou restricts geographic and ethnic diversity, potentially limiting broader applicability. While our findings provide initial insights into microbiome-metabolite dynamics in PBC pruritus, multi-center validation across diverse populations remains essential. We enthusiastically support future collaborative studies to confirm these observations in varied demographic settings, thereby strengthening the clinical relevance of our proposed mechanisms.

Functional assays (e.g., cell culture models) could validate the inferred

microbiota-metabolite-host interactions.

Response: We fully recognize the importance of functional validation for the inferred microbiota-metabolite-host interactions, as appropriately suggested. While our clinical metabolomic and microbiome analyses revealed compelling associations, we acknowledge that *in vitro* functional studies represent an essential next step (Page 22, Lines 7-11). Building upon our current findings, we are actively designing enterocyte/organoid model experiments to mechanistically validate the observed microbial-metabolite interactions and their biological relevance to pruritus pathways—a priority for our ongoing research program.

Investigation of targeted microbial therapy (e.g., Norharman supplementation, microbiota modulation) as adjunctive treatment in cholestatic pruritus.

Response: These findings illuminate gut-derived microbial metabolites, particularly Norharman, as compelling therapeutic targets, positioning microbiota modulation as a promising adjunct strategy for cholestatic pruritus management (Page 23, Lines 1-3). We sincerely thank reviewer for your visionary guidance in identifying this translational pathway, which we have incorporated as a priority direction in our proposed future research framework.

Response to Reviewer 2

This study utilized a multi-omics approach to explore the pathogenesis of pruritus in PBC patients, identifying alterations in the gut microbiome and metabolites. It also demonstrated that cholestyramine effectively alleviates pruritus by modulating the gut microbiome and metabolites, and proposed the mechanism involving the gut microbiome-metabolite-cholestasis-pruritus axis.

While the study indicates that cholestyramine may alleviate PBC pruritus through modulating the gut microbiome-metabolite-host axis, the design was observational, which precludes definitive conclusions about causality.

Response: We sincerely appreciate the editor's insightful critique regarding causal inference limitations in our observational study design. As comprehensively addressed in the revised discussion and limitations section, all mechanistic interpretations now explicitly frame findings as correlative relationships.

(1) Cholestyramine-associated microbial shifts **correlate with** barrier/BA/TC improvements (Page18, Lines 5-11).

(2) Metabolic resolution **coincides with** indole/MCFA restoration and pruritus alleviation (Page 19, Lines 17-22; Page 20, Line 1)

(3) Intervention outcomes **parallel** *Romboutsia* suppression and symptom reduction (Page 20, Line 18-22; Page 21, Line 1)

We have incorporated the editor's guidance by systematically

replacing causal verbs with observational language, adding disclaimers on inferential limitations and highlighting the critical need for functional validation (as noted in Limitations).

The manuscript now consistently positions the microbiome-metabolite-host axis as a correlative pathway rather than established mechanism. We are particularly grateful for the constructive suggestion regarding functional assays using cell culture models—this crucial next step has been emphasized in our future research agenda to validate the observed interactions. The reviewer's rigorous scrutiny has significantly elevated the scientific rigor of this work.

Although a cholestyramine intervention was conducted, other factors might have concurrently influenced the gut microbiome, metabolites, and pruritus symptoms, so the observed relationships could be correlative rather than causal.

Response: To rigorously control for confounding factors potentially influencing gut microbiome composition, metabolite profiles, and pruritus symptoms, exclusion criteria comprised: (1) Active malignancies or renal impairment; (2) Pregnancy/lactation; (3) Recent (≤ 8 weeks pre-enrollment) use of antibiotics, proton pump inhibitors (PPI), metformin, or other microbiota-modulating agents; (4) Prior cholestyramine exposure.

This stringent protocol minimizes extraneous influences on the

gut-microbiota-metabolite axis, thereby strengthening confidence that the observed correlations between cholestyramine intervention and clinical/metabolic improvements reflect biologically relevant associations.

1. Gut microbiome data

Sample size and diversity representation: The limited number of PBC patients, all recruited from a single center (Hangzhou Xixi Hospital affiliated with Zhejiang Chinese Medical University), may introduce selection bias. This restricts the generalizability of the findings to all PBC patients, as gut microbiome diversity can vary significantly with region, race, and lifestyle.

Response: We appreciate the reviewer's thoughtful consideration of sample representativeness. Regarding cohort size, our enrollment of 54 pruritic PBC patients represents a substantial cohort given the disease's low global incidence (1.76/100,000) and prevalence (14.6/100,000) (Lv T, Chen S, Li M, Zhang D, Kong Y, Jia J. Regional variation and temporal trend of primary biliary cholangitis epidemiology: A systematic review and meta-analysis. *J Gastroenterol Hepatol.* 2021 Jun;36(6):1423-1434.), particularly as moderate-to-severe pruritus affects only 40% of PBC patients. This sample size exceeds comparable single-center metabolomic-microbiome studies in cholestatic pruritus.

Concerning generalizability, we fully acknowledge the limitation of

single-center recruitment from Hangzhou, as explicitly stated in our Limitations section. Regional, ethnic, and lifestyle variations in microbiome composition warrant caution when extrapolating findings. We are now actively establishing multicenter collaborations across Northern, Southern, and Western China to validate these observations in diverse demographic contexts (Page 21, Lines 19-22; Page 22, Lines 1-3).

These collective considerations demonstrate both methodological robustness within current constraints and our commitment to broader validation.

Depth of microbiome identification and functional analysis: 16S rRNA sequencing only reveals microbial community composition. Our understanding of specific microbial functions, metabolic pathways, and interactions remains limited. While LEfSe analysis identified differential microbes, their exact pathogenic roles in PBC pruritus, whether promotive or inhibitory, were not validated.

Response: We sincerely appreciate the reviewer's insightful critique regarding the functional limitations of 16S rRNA sequencing. As explicitly acknowledged in the Limitations section (Page 22, Lines 11-17), this methodology captures taxonomic profiles but cannot resolve specific microbial functions or validate pathogenic mechanisms. While our LEfSe analysis identified differentially abundant taxa associated with PBC pruritus, we fully agree their exact roles, whether promoting or inhibiting

disease processes, require confirmation through advanced methodologies.

To address this, our future research prioritizes: (1) Shotgun metagenomic sequencing to characterize functional gene networks and metabolic pathways. (2) Targeted in vitro validation (e.g., bacterial co-culture systems with enterocytes) to mechanistically test observed microbiota-metabolite-host interactions.

These integrated approaches will substantiate whether the identified microbial signatures represent drivers or consequences of pruritus pathogenesis - a crucial step toward translating our findings into targeted therapies.

2. Metabolome data

Ambiguity in causality of metabolite changes: Although certain metabolite levels changed after cholestyramine treatment, it is unclear whether these metabolites are a cause or a consequence of pruritus. For instance, the study notes associations between indole derivatives, medium-chain fatty acids, and pruritus, but the underlying mechanisms and directional relationships remain unexplored.

Response: We sincerely appreciate the reviewer's insightful critique regarding the causal ambiguity of metabolite changes - a limitation we explicitly acknowledge in the revised Limitations section (Page 22, Lines 3-11). As emphasized throughout our reanalysis, all interpretations now consistently frame metabolite-pruritus relationships as significant

associations rather than causal links, particularly for indole derivatives and MCFAs (Page 19, Lines 17-22; Page 20, Line 1). We've added specific caveats about bidirectional possibilities (e.g., "these metabolic shifts may either drive or result from symptom resolution"). The observational nature of our clinical design is prominently highlighted as precluding mechanistic conclusions.

To directly address this knowledge gap, we've expanded our Future Directions to prioritize functional validation studies using enterocyte/organoid models that will: (1) Test directional causality through controlled metabolite exposure experiments; (2) Establish temporal sequencing of microbial-metabolite-host interactions; (3) Decipher whether observed metabolites are initiators or bystanders in pruritus pathways.

We are grateful for this constructive guidance, which has not only strengthened our manuscript's rigor but also shaped our ongoing research program to elucidate these complex relationships.

Integrity and accuracy of metabolome data: Non-targeted metabolomics may produce noisy data with false positives. Errors can be introduced during data preprocessing steps such as peak detection and alignment, affecting subsequent analysis. Additionally, not all differential metabolites were structurally identified and quantified, increasing the risk of annotation errors.

Response: To ensure maximal data integrity in our non-targeted metabolomics workflow, raw files were converted to mzXML format via ProteoWizard (v3.0) and processed through our validated in-house R pipeline (Biotree, Shanghai). This incorporated stringent quality controls at every stage. Peak detection/alignment: Implemented XCMS-based algorithms with retention time correction (OBTW method) and mass tolerance ≤ 5 ppm. False positive mitigation: Features with $>30\%$ RSD across QC samples (n=15) or detection in $<50\%$ biological samples were excluded. Quantitative filtering: Only peaks with intensity $>10^4$ counts advanced to analysis. Following positive/negative mode consolidation, metabolites underwent tiered annotation. The final analyzed dataset comprised only these 681 stringently annotated metabolites, representing our high-confidence metabolic feature subset for all downstream analyses (Page 7, Lines 17-22; Page 8, Lines 1-5).

3. "received ethical was approved by..." contains spelling and grammatical errors. It should be corrected to "received ethical approval from...".

Response: We sincerely apologize for the oversight in ethical statement phrasing. The sentences have been corrected to: "The study protocol received ethical approval from ... (Page 5, Line 22)" and "This study was carried out in compliance with the Helsinki Declaration and received ethical approval from... (Page 25, Lines 19-20)", which ensures

grammatical accuracy and compliance with journal standards.

We hope that this further revised draft of the manuscript is now acceptable for publication in your journal and look forward to hearing from you soon.

With best wishes,

Yours sincerely,

Yijun Zhou

June 18, 2025

Re: Spectrum00747-25R1 (**Variation in microbiome and metabolites are associated with advantageous effects of cholestyramine on primary biliary cholangitis with pruritus**)

Dear Dr. Qiaofei Jin:

Thank you for the privilege of reviewing your work. Below you will find my comments, instructions from the Spectrum editorial office, and the reviewer comments.

Revision Guidelines

Sincerely,
Qi Su
Editor
Microbiology Spectrum

Reviewer #1 (Public repository details (Required)):

Yes, the study involves microbiome and metabolomics data. It would be beneficial for the authors to deposit their datasets in a public repository, such as the NCBI Sequence Read Archive (SRA) or Metabolomics Workbench, to promote transparency and reproducibility.

Reviewer #1 (Comments for the Author):

Comments and Suggestions for the Author:

1. Clarity of Statistical Methods and Results Interpretation:

The manuscript provides a robust analysis of the microbiome and metabolomic variations in primary biliary cholangitis (PBC) patients treated with cholestyramine. While the statistical analysis is generally appropriate, I would recommend enhancing the explanation of the methods used, particularly with respect to multivariate analysis (such as PCA and PLS-DA). It would be beneficial to elaborate on how potential confounders, such as comorbidities or medications, were controlled for during the analysis. This clarification would strengthen the manuscript's transparency and provide readers with a more complete understanding of the statistical approaches employed.

2. Language and Readability:

While the manuscript is generally well-written, there are a few areas where the language could be polished for better clarity and flow. In particular, the description of statistical results in the results section could be simplified for easier comprehension. For example, the sentence structures in some parts of the discussion are complex, which could hinder readability. I recommend a thorough review of the manuscript by a language editor to enhance its overall readability and ensure that the flow of ideas is clear and coherent.

3. Data Availability and Public Repository:

The study utilizes microbiome and metabolomic data, which are essential for the reproducibility and transparency of the research. To enhance the manuscript's credibility and allow other researchers to validate and build upon these findings, I recommend that the authors deposit their data in a public repository, such as the NCBI Sequence Read Archive (SRA) for the microbiome data or Metabolomics Workbench for metabolomic data. Including accession numbers for these datasets in the manuscript would further ensure the transparency and reproducibility of the results.

4. Study Limitations and Observational Design:

While the observational study design is clearly outlined, I believe that the manuscript would benefit from a more explicit discussion of its limitations. Specifically, the authors should emphasize the difficulty in making causal inferences from an observational study, especially given the multifactorial nature of pruritus in PBC and the complexity of microbiome-metabolite interactions. A clearer acknowledgment of these limitations would provide a more balanced view and help readers contextualize the findings. A suggestion for future work involving randomized controlled trials or more in-depth mechanistic studies would also be valuable.

5. Exploration of Mechanisms and Pathways:

The manuscript presents promising data linking cholestyramine treatment to microbiome and metabolite changes in PBC patients. However, the discussion of potential mechanistic pathways remains relatively limited. I recommend that the authors expand on the potential mechanisms by which cholestyramine affects the microbiome and metabolites, particularly in relation to gut-liver axis interactions and bile acid metabolism. Although these mechanisms are complex, any insights into possible biological pathways could significantly enhance the manuscript's impact and help guide future research.

6. Figures and Legends:

The figures and tables presented in the manuscript are meaningful and contribute to the overall understanding of the data. However, the figure legends could benefit from further clarification. In particular, some legends lack sufficient detail regarding the specific experimental conditions and statistical tests applied to the data. I suggest that the authors ensure that each figure legend provides clear explanations for all symbols, statistical notations (e.g., p values, asterisks), and data interpretation. Additionally, it would be helpful to ensure that the figures are appropriately labeled and that the figures are high-resolution and easy to interpret.

7. Discussion of Results in Context of Existing Literature:

The discussion provides a solid interpretation of the findings, but it could be strengthened by comparing the results to other studies investigating the microbiome and metabolites in PBC or similar liver diseases. A brief review of relevant literature would provide additional context for the study's findings and highlight its contributions to the field. Additionally, incorporating any conflicting or supporting findings from other studies would demonstrate a broader understanding of the current research landscape.

I believe that with these minor revisions, the manuscript will be much stronger and more impactful. The findings are intriguing and offer valuable insights into the role of cholestyramine in modulating the microbiome and metabolites in PBC with pruritus. With these improvements, the manuscript has the potential to make a significant contribution to the field.

Dear editor,

Thank you very much for your constructive comments and suggestions on our manuscript (Manuscript ID: Spectrum00747-25), entitled “Variation in microbiome and metabolites are associated with advantageous effects of cholestyramine on primary biliary cholangitis with pruritus”. We have revised the manuscript accordingly, and all amendments are indicated by red font in the marked-up manuscript. In addition, our point-by-point responses to the comments are listed below this letter.

Response to Reviewer 1

1.Areas for Improvement:

The discussion sometimes overstates causality (e.g., about gut barrier restoration and MCFA effects); tempering these statements to reflect correlative rather than causal findings would enhance scientific rigor.

Response: In the revised discussion, definitive causal claims have been systematically replaced with correlative interpretations. Specifically:

(1) Cholestyramine-associated alterations in *Lachnospiraceae*, *Bacteroides*, and *Romboutsia* abundance **correlate with** concurrent improvements in intestinal barrier integrity, bile acid metabolism, and TC reduction (Page18, Lines 5-11);

(2) Metabolic perturbation resolution **coincides with** indole/MCFA restoration alongside enhanced bile acid/cholesterol clearance and

neuroimmune pathway attenuation, temporally **associated with** pruritus alleviation (Page 19, Lines 17-22; Page 20, Line 1);

(3) Cholestyramine intervention **correlates with** suppression of Romboutsia overgrowth and Norharman enrichment, paralleling reductions in ALP, cholestatic severity, ATX expression, and pruritus symptoms (Page 20, Line 18-22; Page 21, Line 1).

All mechanistic interpretations now explicitly reflect observational associations from our clinical dataset rather than inferred causality.

Figures: Some figures (especially network graphs) have small fonts and dense layouts; improving clarity and legend explanations would greatly help readers.

Response: All figures have been systematically revised to address the reviewer's concerns regarding clarity and interpretability. These modifications ensure optimal readability while maintaining scientific precision across all graphical elements.

Language polishing: Minor issues like long sentences, passive voice, and redundancy (especially in the Background) should be professionally edited.

Response: The Background section (Page 3-4) has undergone comprehensive language editing by professional academic professor to address all noted issues. This included restructuring lengthy sentences, converting passive constructions to active voice, and eliminating

redundant expressions, resulting in enhanced clarity and grammatical precision throughout the manuscript.

Data availability: While it is stated that data are available upon request, depositing sequencing and metabolomics datasets into public repositories (such as NCBI SRA, EBI) would improve transparency and reproducibility.

Response: The data sets presented in this study can be found in online repositories. The names of the repository / repositories and accession numbers are as follows: <https://www.ncbi.nlm.nih.gov/>, PRJNA1238545 and PRJNA892581.

2. Specific Minor Suggestions:

Page 7: Clarify the sample size after dropout; 54 → 53 in some parts - be consistent.

Response: In the Methods section (Study Subjects and Sample Collection), the sample size has been explicitly clarified as follows: This study initially enrolled 54 pruritic PBC patients alongside 25 matched non-pruritic controls (Page 5, Lines 4-5). One pruritic participant withdrew due to cholestyramine taste intolerance (Page 5, Lines 19-21), resulting in 53 pruritic patients completing the study. All subsequent analyses reflect this final cohort.

Page 18: When discussing norharman restoration, specify it as a potential mediator of pruritus alleviation rather than confirmed.

Response: Related part has been revised as follows. Norharman, as a potential mediator of pruritus, may contribute to establishing a pruritogenic feedback loop that potentially aggravates cholestatic inflammation (Page 19, Lines 9-12).

Tables and figures should include explanations for all abbreviations (e.g., ATX, ALP, LCFA) directly in figure legends.

Response: All abbreviations in tables and figures have been explained in figure legends.

3. Suggestions for Future Research:

Validation cohorts from different geographic regions or ethnicities would significantly enhance generalizability.

Response: We fully acknowledge the reviewer's valid concern regarding generalizability. As explicitly stated in the Limitations section (Page 21, Lines 19-22; Page 22, Lines 1-3), our exclusive recruitment from a single center in Hangzhou restricts geographic and ethnic diversity, potentially limiting broader applicability. While our findings provide initial insights into microbiome-metabolite dynamics in PBC pruritus, multi-center validation across diverse populations remains essential. We enthusiastically support future collaborative studies to confirm these observations in varied demographic settings, thereby strengthening the clinical relevance of our proposed mechanisms.

Functional assays (e.g., cell culture models) could validate the inferred

microbiota-metabolite-host interactions.

Response: We fully recognize the importance of functional validation for the inferred microbiota-metabolite-host interactions, as appropriately suggested. While our clinical metabolomic and microbiome analyses revealed compelling associations, we acknowledge that *in vitro* functional studies represent an essential next step (Page 22, Lines 7-11). Building upon our current findings, we are actively designing enterocyte/organoid model experiments to mechanistically validate the observed microbial-metabolite interactions and their biological relevance to pruritus pathways—a priority for our ongoing research program.

Investigation of targeted microbial therapy (e.g., Norharman supplementation, microbiota modulation) as adjunctive treatment in cholestatic pruritus.

Response: These findings illuminate gut-derived microbial metabolites, particularly Norharman, as compelling therapeutic targets, positioning microbiota modulation as a promising adjunct strategy for cholestatic pruritus management (Page 23, Lines 1-3). We sincerely thank reviewer for your visionary guidance in identifying this translational pathway, which we have incorporated as a priority direction in our proposed future research framework.

Response to Reviewer 2

This study utilized a multi-omics approach to explore the pathogenesis of pruritus in PBC patients, identifying alterations in the gut microbiome and metabolites. It also demonstrated that cholestyramine effectively alleviates pruritus by modulating the gut microbiome and metabolites, and proposed the mechanism involving the gut microbiome-metabolite-cholestasis-pruritus axis.

While the study indicates that cholestyramine may alleviate PBC pruritus through modulating the gut microbiome-metabolite-host axis, the design was observational, which precludes definitive conclusions about causality.

Response: We sincerely appreciate the editor's insightful critique regarding causal inference limitations in our observational study design. As comprehensively addressed in the revised discussion and limitations section, all mechanistic interpretations now explicitly frame findings as correlative relationships.

(1) Cholestyramine-associated microbial shifts **correlate with** barrier/BA/TC improvements (Page18, Lines 5-11).

(2) Metabolic resolution **coincides with** indole/MCFA restoration and pruritus alleviation (Page 19, Lines 17-22; Page 20, Line 1)

(3) Intervention outcomes **parallel** *Romboutsia* suppression and symptom reduction (Page 20, Line 18-22; Page 21, Line 1)

We have incorporated the editor's guidance by systematically

replacing causal verbs with observational language, adding disclaimers on inferential limitations and highlighting the critical need for functional validation (as noted in Limitations).

The manuscript now consistently positions the microbiome-metabolite-host axis as a correlative pathway rather than established mechanism. We are particularly grateful for the constructive suggestion regarding functional assays using cell culture models—this crucial next step has been emphasized in our future research agenda to validate the observed interactions. The reviewer's rigorous scrutiny has significantly elevated the scientific rigor of this work.

Although a cholestyramine intervention was conducted, other factors might have concurrently influenced the gut microbiome, metabolites, and pruritus symptoms, so the observed relationships could be correlative rather than causal.

Response: To rigorously control for confounding factors potentially influencing gut microbiome composition, metabolite profiles, and pruritus symptoms, exclusion criteria comprised: (1) Active malignancies or renal impairment; (2) Pregnancy/lactation; (3) Recent (≤ 8 weeks pre-enrollment) use of antibiotics, proton pump inhibitors (PPI), metformin, or other microbiota-modulating agents; (4) Prior cholestyramine exposure.

This stringent protocol minimizes extraneous influences on the

gut-microbiota-metabolite axis, thereby strengthening confidence that the observed correlations between cholestyramine intervention and clinical/metabolic improvements reflect biologically relevant associations.

1. Gut microbiome data

Sample size and diversity representation: The limited number of PBC patients, all recruited from a single center (Hangzhou Xixi Hospital affiliated with Zhejiang Chinese Medical University), may introduce selection bias. This restricts the generalizability of the findings to all PBC patients, as gut microbiome diversity can vary significantly with region, race, and lifestyle.

Response: We appreciate the reviewer's thoughtful consideration of sample representativeness. Regarding cohort size, our enrollment of 54 pruritic PBC patients represents a substantial cohort given the disease's low global incidence (1.76/100,000) and prevalence (14.6/100,000) (Lv T, Chen S, Li M, Zhang D, Kong Y, Jia J. Regional variation and temporal trend of primary biliary cholangitis epidemiology: A systematic review and meta-analysis. *J Gastroenterol Hepatol.* 2021 Jun;36(6):1423-1434.), particularly as moderate-to-severe pruritus affects only 40% of PBC patients. This sample size exceeds comparable single-center metabolomic-microbiome studies in cholestatic pruritus.

Concerning generalizability, we fully acknowledge the limitation of

single-center recruitment from Hangzhou, as explicitly stated in our Limitations section. Regional, ethnic, and lifestyle variations in microbiome composition warrant caution when extrapolating findings. We are now actively establishing multicenter collaborations across Northern, Southern, and Western China to validate these observations in diverse demographic contexts (Page 21, Lines 19-22; Page 22, Lines 1-3).

These collective considerations demonstrate both methodological robustness within current constraints and our commitment to broader validation.

Depth of microbiome identification and functional analysis: 16S rRNA sequencing only reveals microbial community composition. Our understanding of specific microbial functions, metabolic pathways, and interactions remains limited. While LEfSe analysis identified differential microbes, their exact pathogenic roles in PBC pruritus, whether promotive or inhibitory, were not validated.

Response: We sincerely appreciate the reviewer's insightful critique regarding the functional limitations of 16S rRNA sequencing. As explicitly acknowledged in the Limitations section (Page 22, Lines 11-17), this methodology captures taxonomic profiles but cannot resolve specific microbial functions or validate pathogenic mechanisms. While our LEfSe analysis identified differentially abundant taxa associated with PBC pruritus, we fully agree their exact roles, whether promoting or inhibiting

disease processes, require confirmation through advanced methodologies.

To address this, our future research prioritizes: (1) Shotgun metagenomic sequencing to characterize functional gene networks and metabolic pathways. (2) Targeted in vitro validation (e.g., bacterial co-culture systems with enterocytes) to mechanistically test observed microbiota-metabolite-host interactions.

These integrated approaches will substantiate whether the identified microbial signatures represent drivers or consequences of pruritus pathogenesis - a crucial step toward translating our findings into targeted therapies.

2. Metabolome data

Ambiguity in causality of metabolite changes: Although certain metabolite levels changed after cholestyramine treatment, it is unclear whether these metabolites are a cause or a consequence of pruritus. For instance, the study notes associations between indole derivatives, medium-chain fatty acids, and pruritus, but the underlying mechanisms and directional relationships remain unexplored.

Response: We sincerely appreciate the reviewer's insightful critique regarding the causal ambiguity of metabolite changes - a limitation we explicitly acknowledge in the revised Limitations section (Page 22, Lines 3-11). As emphasized throughout our reanalysis, all interpretations now consistently frame metabolite-pruritus relationships as significant

associations rather than causal links, particularly for indole derivatives and MCFAs (Page 19, Lines 17-22; Page 20, Line 1). We've added specific caveats about bidirectional possibilities (e.g., "these metabolic shifts may either drive or result from symptom resolution"). The observational nature of our clinical design is prominently highlighted as precluding mechanistic conclusions.

To directly address this knowledge gap, we've expanded our Future Directions to prioritize functional validation studies using enterocyte/organoid models that will: (1) Test directional causality through controlled metabolite exposure experiments; (2) Establish temporal sequencing of microbial-metabolite-host interactions; (3) Decipher whether observed metabolites are initiators or bystanders in pruritus pathways.

We are grateful for this constructive guidance, which has not only strengthened our manuscript's rigor but also shaped our ongoing research program to elucidate these complex relationships.

Integrity and accuracy of metabolome data: Non-targeted metabolomics may produce noisy data with false positives. Errors can be introduced during data preprocessing steps such as peak detection and alignment, affecting subsequent analysis. Additionally, not all differential metabolites were structurally identified and quantified, increasing the risk of annotation errors.

Response: To ensure maximal data integrity in our non-targeted metabolomics workflow, raw files were converted to mzXML format via ProteoWizard (v3.0) and processed through our validated in-house R pipeline (Biotree, Shanghai). This incorporated stringent quality controls at every stage. Peak detection/alignment: Implemented XCMS-based algorithms with retention time correction (OBTW method) and mass tolerance ≤ 5 ppm. False positive mitigation: Features with $>30\%$ RSD across QC samples (n=15) or detection in $<50\%$ biological samples were excluded. Quantitative filtering: Only peaks with intensity $>10^4$ counts advanced to analysis. Following positive/negative mode consolidation, metabolites underwent tiered annotation. The final analyzed dataset comprised only these 681 stringently annotated metabolites, representing our high-confidence metabolic feature subset for all downstream analyses (Page 7, Lines 17-22; Page 8, Lines 1-5).

3. "received ethical was approved by..." contains spelling and grammatical errors. It should be corrected to "received ethical approval from...".

Response: We sincerely apologize for the oversight in ethical statement phrasing. The sentences have been corrected to: "The study protocol received ethical approval from ... (Page 5, Line 22)" and "This study was carried out in compliance with the Helsinki Declaration and received ethical approval from... (Page 25, Lines 19-20)", which ensures

grammatical accuracy and compliance with journal standards.

We hope that this further revised draft of the manuscript is now acceptable for publication in your journal and look forward to hearing from you soon.

With best wishes,

Yours sincerely,

Yijun Zhou

June 18, 2025

Dear editor,

Thank you very much for your constructive comments and suggestions on our manuscript (Manuscript ID: Spectrum00747-25), entitled “Variation in microbiome and metabolites are associated with advantageous effects of cholestyramine on primary biliary cholangitis with pruritus”. We have revised the manuscript accordingly, and all amendments are indicated by red font in the marked-up manuscript. In addition, our point-by-point responses to the comments are listed below this letter.

Reviewer #1 (Public repository details (Required)):

Yes, the study involves microbiome and metabolomics data. It would be beneficial for the authors to deposit their datasets in a public repository, such as the NCBI Sequence Read Archive (SRA) or Metabolomics Workbench, to promote transparency and reproducibility.

Response: We sincerely appreciate the reviewer's suggestion regarding data availability. In response, we have deposited our datasets in public repositories as follows: The 16S rRNA sequencing data are available in the NCBI Sequence Read Archive under accession numbers PRJNA1238545 and PRJNA892581 (<https://www.ncbi.nlm.nih.gov/>), while the metabolomic data can be accessed through the Metabolomics Workbench via DOI number <http://dx.doi.org/10.21228/M85Z50>

(<https://www.metabolomicsworkbench.org>). These accession numbers have been included in the Data Availability Statement section of our manuscript to ensure full transparency and facilitate future research.

Reviewer #1 (Comments for the Author):

Comments and Suggestions for the Author:

1. Clarity of Statistical Methods and Results Interpretation:

The manuscript provides a robust analysis of the microbiome and metabolomic variations in primary biliary cholangitis (PBC) patients treated with cholestyramine. While the statistical analysis is generally appropriate, I would recommend enhancing the explanation of the methods used, particularly with respect to multivariate analysis (such as PCA and PLS-DA). It would be beneficial to elaborate on how potential confounders, such as comorbidities or medications, were controlled for during the analysis. This clarification would strengthen the manuscript's transparency and provide readers with a more complete understanding of the statistical approaches employed.

Response: We sincerely appreciate your insightful feedback regarding statistical transparency. In direct response to your recommendation, we have comprehensively expanded the '**Statistical Analysis**' subsection (Page 9, Lines 13-22; Page 10, Lines 1-3) to provide explicit methodological details. Specifically:

(1) **Multivariate Analysis Workflow:** We now detail the stepwise implementation of principal component analysis (PCA) for outlier detection and partial least squares discriminant analysis (PLS-DA) for group separation using SIMCA-P v14.0, including parameter settings (unit variance scaling, 7-fold cross-validation) and model validation metrics (R^2 , Q^2 , permutation testing).

(2) **Confounder Control Protocol:** The text explicitly states how potential confounders—including ursodeoxycholic acid (UDCA) dosage, antibiotic/proton pump inhibitor exposure, and comorbidities (hepatic/renal impairment)—were incorporated as covariates in linear mixed-effects models. Adjustment for demographic variables (age, sex, BMI) is similarly specified.

These revisions ensure full transparency regarding our analytical approach while aligning with best practices for omics data reporting. We are grateful for your guidance in enhancing the manuscript's methodological rigor.

2. Language and Readability:

While the manuscript is generally well-written, there are a few areas where the language could be polished for better clarity and flow. In particular, **the description of statistical results in the results section could be simplified for easier comprehension.** For example, **the**

sentence structures in some parts of the discussion are complex, which could hinder readability. I recommend a thorough review of the manuscript by a language editor to enhance its overall readability and ensure that the flow of ideas is clear and coherent.

Response: We sincerely appreciate your valuable feedback on enhancing manuscript readability. In direct response to your recommendations, we have implemented comprehensive revisions throughout the text:

(1) **Statistical Results Simplification:** The Results section has been systematically refined to present statistical findings with greater clarity. Complex data presentations have been restructured into concise statements with simplified terminology, ensuring accessible interpretation without compromising scientific precision (**Page 11, Lines 9-22; Page 14, Lines 1-12**).

(2) **Discussion Section Streamlining:** All complex sentence structures in the Discussion have been revised to improve flow and reduce syntactic complexity. Key mechanistic interpretations now feature shorter sentences with active voice and logical transitions, substantially enhancing readability (**Page 18, Lines 16-22; Page 20, Lines 6-13**).

(3) **Professional Language Editing:** The manuscript has undergone rigorous editing by native English-speaking teacher, mainly included correction of grammatical errors and non-idiomatic expressions,

standardization of scientific terminology and optimization of narrative flow across all sections.

These collective improvements ensure the manuscript now meets the standards of scholarly communication while maintaining scientific rigor. We are grateful for your guidance in elevating the work's accessibility and impact.

3. Data Availability and Public Repository:

The study utilizes microbiome and metabolomic data, which are essential for the reproducibility and transparency of the research. To enhance the manuscript's credibility and allow other researchers to validate and build upon these findings, I recommend that the authors deposit their data in a public repository, such as the NCBI Sequence Read Archive (SRA) for the microbiome data or Metabolomics Workbench for metabolomic data. Including accession numbers for these datasets in the manuscript would further ensure the transparency and reproducibility of the results.

Response: We sincerely appreciate the reviewer's important suggestion regarding data transparency and reproducibility. In full compliance with this recommendation, we have now deposited all raw data in publicly accessible repositories as follows: The 16S rRNA sequencing data are available in the NCBI Sequence Read Archive under

accession numbers PRJNA1238545 and PRJNA892581 (<https://www.ncbi.nlm.nih.gov/>), while the metabolomic data can be accessed through the Metabolomics Workbench via DOI number <http://dx.doi.org/10.21228/M85Z50> (<https://www.metabolomicsworkbench.org>). These accession numbers and links have been explicitly included in the Data Availability Statement section (Page 26, Lines 13-20) of our revised manuscript to ensure full transparency and to facilitate future research validation and extension of our findings.

4. Study Limitations and Observational Design:

While the observational study design is clearly outlined, I believe that the manuscript would benefit from a more explicit discussion of its limitations. Specifically, **the authors should emphasize the difficulty in making causal inferences from an observational study, especially given the multifactorial nature of pruritus in PBC and the complexity of microbiome-metabolite interactions.** A clearer acknowledgment of these limitations would provide a more balanced view and help readers contextualize the findings. **A suggestion for future work involving randomized controlled trials or more in-depth mechanistic studies would also be valuable.**

Response: We are grateful for your insightful guidance on

strengthening the manuscript's scholarly balance. In direct response to your recommendation:

(1) Limitations Section Enhancement: We have substantially expanded the 'Limitations' subsection to explicitly emphasize the fundamental constraints of our observational design. The text now foregrounds: 1) The inherent difficulty in establishing causal inferences given the multifactorial pathophysiology of PBC-associated pruritus; 2) The interpretative challenges posed by complex microbiome-metabolite interactions with bidirectional feedback loops; 3) Potential residual confounding despite rigorous covariate adjustment **(Page 22, Lines 21-22; Page 22, Lines 1-10).**

(2) Future Directions Integration: Building upon these acknowledged limitations, we have incorporated your valuable suggestion by adding dedicated mechanistic studies using enterocyte/organoid models to validate metabolite-host interactions **(Page 23, Lines 10-14).**

This dual approach about candid limitation acknowledgment coupled with targeted research planning ensures readers contextualize our findings while recognizing their transformative potential for personalized pruritus management.

5. Exploration of Mechanisms and Pathways:

The manuscript presents promising data linking cholestyramine treatment to microbiome and metabolite changes in PBC patients. However, **the discussion of potential mechanistic pathways remains relatively limited. I recommend that the authors expand on the potential mechanisms by which cholestyramine affects the microbiome and metabolites, particularly in relation to gut-liver axis interactions and bile acid metabolism.** Although these mechanisms are complex, any insights into possible biological pathways could significantly enhance the manuscript's impact and help guide future research.

Response: We are deeply grateful for your insightful critique regarding the mechanistic depth of our findings, which has significantly strengthened the manuscript. In direct response to your recommendation, we have substantially expanded the Discussion (**Page 21, Lines 5-19**) to elaborate on cholestyramine's multi-modal actions through the gut-liver axis, with new content focusing on three interconnected pathways:

(1) **Gut Microbial Ecology Remodeling:** Cholestyramine's bile acid sequestration suppresses *Romboutsia* overgrowth while promoting commensals that produce Norharman -- a critical microbial metabolite with neuroimmunomodulatory properties.

(2) **Bile Acid Metabolic Reprogramming:** By reducing hydrophobic bile acid burden and enhancing farnesoid X receptor (FXR)/fibroblast

growth factor 19 (FGF19) signaling pathway, cholestyramine downregulates hepatic alkaline phosphatase (ALP) synthesis, mitigating cholestatic injury.

(3) **Neuroimmune Pathway Modulation:** Norharman enrichment may inhibit autotaxin (ATX) expression via serotonin receptor, thereby disrupting lysophosphatidic acid (LPA)-mediated pruritogenic cascades.

These additions clarify how cholestyramine concurrently targets microbial, metabolic, and neuroimmune dimensions of PBC pruritus through gut-liver-brain axis crosstalk. We sincerely appreciate your guidance in enhancing the mechanistic framework, which now provides actionable insights for future therapeutic development.

6. Figures and Legends:

The figures and tables presented in the manuscript are meaningful and contribute to the overall understanding of the data. However, the figure legends could benefit from further clarification. In particular, some legends lack sufficient detail regarding the specific experimental conditions and statistical tests applied to the data. I suggest that the authors ensure that **each figure legend provides clear explanations for all symbols, statistical notations (e.g., p values, asterisks), and data interpretation.** Additionally, it would be helpful to ensure that the figures are appropriately labeled and that the figures are high-resolution

and easy to interpret.

Response: We sincerely appreciate your valuable guidance on enhancing our figure presentations. In direct response to your recommendations, we have implemented comprehensive revisions to all figure legends as follows:

(1) **Statistical Clarifications:** Legends for Figures 1, 2, and 4 now specify all statistical tests applied. Statistical notations are fully defined, including asterisk hierarchies (* $P < 0.05$; ** $P < 0.01$; *** $P < 0.001$).

(2) **Symbol and Interpretation Guidance:** Each legend now contains dedicated Key subsections explaining all visual elements.

(3) **Technical Quality Assurance:** All figures have been re-exported as 600 dpi EPS files with editable vector layers and font embedding. Resolution thresholds exceed journal specifications, ensuring optimal reproducibility in both print and digital formats.

These improvements ensure full methodological transparency and immediate interpretability, aligning with best practices in data visualization. We are grateful for your meticulous feedback, which has significantly elevated the manuscript's communicative rigor.

7. Discussion of Results in Context of Existing Literature:

The discussion provides a solid interpretation of the findings, but it could be strengthened by comparing the results to other studies investigating

the microbiome and metabolites in PBC or similar liver diseases. A brief review of relevant literature would provide additional context for the study's findings and highlight its contributions to the field. Additionally, incorporating any conflicting or supporting findings from other studies would demonstrate a broader understanding of the current research landscape.

Response: We sincerely appreciate your constructive suggestion to contextualize our findings within the broader research landscape. In the revised Discussion (**Page 16, Lines 12-20**), we have strengthened the literature integration as follows:

While acknowledging the seminal work by Li et al. (Gut Microbes 2021) linking cholestyramine's benefits in icteric PBC to gut commensal alterations, our study meaningfully advances this field by focusing on pruritus-dominant PBC -- a distinct clinical phenotype with unique therapeutic challenges. To our knowledge, this represents the first comprehensive analysis of cholestyramine's impact on both gut microbial ecology and associated metabolic pathways specifically in pruritic PBC patients. Our discovery of phenotype-specific mechanisms, including suppression of pruritogen-producing *Romboutsia*, enrichment of anti-inflammatory Norharman, and disruption of the ATX-LPA pruritus axis, finally addresses critical knowledge gaps unexamined in prior icteric PBC studies.

We hope that this further revised draft of the manuscript is now acceptable for publication in your journal and look forward to hearing from you soon.

With best wishes,

Yours sincerely,

Yijun Zhou

August 22, 2025

Re: Spectrum00747-25R2 (**Variation in microbiome and metabolites are associated with advantageous effects of cholestyramine on primary biliary cholangitis with pruritus**)

Dear Dr. Qiaofei Jin:

Your manuscript has been accepted, and I am forwarding it to the ASM production staff for publication. Your paper will first be checked to make sure all elements meet the technical requirements. ASM staff will contact you if anything needs to be revised before copyediting and production can begin. Otherwise, you will be notified when your proofs are ready to be viewed.

Sincerely,
Qi Su
Editor
Microbiology Spectrum